# Public participation in healthcare safety: A tripartite evolutionary game model with evidence from diverse international cases

ZhiQiang Zeng[1,2], Saratha Sathasivam[2]*, Jing Xin[3], Huan Zhao[4]

1 School of Computer Information Engineering, Nanchang Institute of Technology, Nanchang, Jiangxi, China, 2 School of Mathematical Sciences, Universiti Sains Malaysia, USM, Penang, Malaysia, 3 Department of Respiratory Medicine, Jiangxi Provincial Chest Hospital, Nanchang, Jiangxi, China, 4 Institute of Innovation and Entrepreneurship, Nanchang Institute of Technology, Nanchang, Jiangxi, China

* saratha@usm.my

## Abstract

As healthcare systems grow in complexity, ensuring medical safety requires moving beyond traditional top-down regulation. While public participation is increasingly recognized as a vital component, a robust, generalizable framework to guide its implementation has been lacking. This study addresses this critical gap by proposing and rigorously validating a tripartite evolutionary game model that integrates the public, medical institutions, and government authorities. We test the model's universality and effectiveness against empirical data from three diverse international case studies: tuberculosis (TB) treatment adherence in Saudi Arabia, COVID-19 vaccination compliance in China, and antibiotic prescription supervision in Vietnam. Our analysis reveals that higher health risks and public exposure rates act as powerful catalysts, significantly enhancing participation from both the public and institutions, thereby accelerating the system's convergence to a stable, compliant state. We also find that medical institutions are highly sensitive to penalty intensity, adopting compliant practices only when a critical threshold is surpassed. These findings confirm the existence of a "virtuous cycle," where engaged citizens and stringent oversight collaboratively improve medical compliance, a mechanism that holds true across varied healthcare contexts. This research provides not only a validated theoretical framework but also actionable insights for policymakers to design evidence-based, participatory regulatory strategies that can enhance the resilience and sustainability of global healthcare systems.

## 1. Introduction

With the rapid growth of economic development and the increasing diversity of medical needs, intelligent medical systems have brought significant changes and

**Data availability statement:** Data Availability Statement: This study utilized both previously published data and newly incorporated datasets to support findings. Tuberculosis (TB) data: We did not directly use raw individual patient data but utilized aggregated summary statistics and key parameter values published by AlSahafi et al. (2019), publicly available through BMC Public Health at https://doi.org/10.1186/s12889-019-7520-8. The original raw dataset used in AlSahafi et al. (2019) is subject to data-sharing restrictions due to privacy concerns but can be obtained from the corresponding author upon reasonable request (email: hassanbinusman@hotmail.com). Guangdong COVID-19 vaccination supervision data: The research data includes aggregated supervision records from the Guangdong Provincial Health Commission (GPHC)'s 2023 COVID-19 Vaccination Supervision Annual Report (publicly accessible at http://wsjkw.gd.gov.cn/) and a cross-sectional survey of 500 Guangdong residents (available at https://doi.org/10.3969/j.issn.1674-2982.2024.02.008). Vietnam antibiotic resistance supervision data: The research data includes: (1) WHO's 2023 Global Antimicrobial Resistance Surveillance Report (focused on Vietnam's rural health-care sector, publicly available at https://www.who.int/); (2) A longitudinal study of 30 rural clinics in Vietnam (accessible at https://doi.org/10.1016/j.apm.2023.06.014).

**Funding:** Funder: The Ministry of Higher Education (MOHE) of Malaysia Grant Scheme: Fundamental Research Grant Scheme (FRGS) Grant Number: FRGS/1/2022/STG06/USM/02/11 Recipient Institution: Universiti Sains Malaysia (USM) We confirm that the funders had no role in the study design, data collection and analysis, decision to publish, or preparation of the manuscript.

**Competing interests:** The author has declared that there is no competing interest.

progress to the medical industry. These systems utilize advanced technology and data analysis to efficiently allocate medical resources and enable continuous real-time health monitoring. The World Health Organization (WHO) has released the "Ten-Year Plan for Global Health Action," advocating for countries to take proactive measures to ensure that medical institutions, medical practices, and public participation jointly achieve medical safety goals through an integrated "health system" approach. Previous studies have explored various aspects of intelligent medical systems and game theory applications in healthcare. For instance, Barrett [1] analyzed the success and failure of international disease eradication programs, while Biaou et al. [2] proposed game theory-based methods to solve the free-rider problem in peer-to-peer networks. Ahmed and Eswarappa [3] used game theory to predict the presence of pathogenic bacteria in both intracellular and extracellular regions during persistent infections. More recently, Deng et al. [4] proposed a blockchain-based dynamic trust access control game mechanism to address issues in open network environments. Li et al. [5] created a model for predicting infectious diseases by using evolutionary game theory and network structure. These studies highlight the potential of game theory in addressing complex healthcare challenges. Martcheva et al. [6] further integrated evolutionary game theory with economic perspectives to analyze how social-distancing policies influence infectious disease dynamics, revealing that individual behavior and policy intensity jointly determine the stability of disease control systems. These studies highlight the potential of game theory in addressing complex healthcare challenges, particularly in public health emergency management. However, existing research often focuses on individual aspects of healthcare systems, such as disease prediction or network security, with limited attention to the role of public participation in medical safety risk supervision. This gap is crucial, as public involvement can significantly enhance the effectiveness of healthcare policies and practices. Therefore, this study aims to fill this gap by developing a tripartite evolutionary game model that incorporates the public, medical institutions, and regulatory authorities. We analyze the interactions and decision-making processes among these stakeholders in the context of medical safety risk supervision. Our study provides a novel perspective on how public participation can influence the stability and effectiveness of healthcare systems, contributing to the sustainable development of intelligent medical services.

[1] explores the reasons for the success and failure of international disease eradication programs [2], Game theory-based methods, specifically the modified Ayo game, are studied to solve the free-rider problem in peer-to-peer networks (P2P) [3], game theory was employed to analyze the whereabouts of pathogenic bacteria throughout persistent infection. Their findings put forth a theoretical model that predicts the presence of these bacteria in both the intracellular and extracellular regions of the host during this type of infection. [7] they investigated the factors related to TB treatment default in Khartoum State, Sudan through a case-control study [8], introduced a novel predictive modeling technique that utilizes coalition game theory and information theory [9]. A network attack and defense evolutionary game decision-making method based on regret minimization algorithm is proposed to solve the limitations of current network security attack and defense decision-making methods in practical applications

assuming that participants are completely rational [10]. Analyzed possible conflicts of interest in contact investigations through game theory [11], developed a game theory model to examine how people make choices regarding vaccination and clean water consumption during a cholera outbreak [12]. Examined the influence of knowledge spillover from medical big data on patients' medical decisions within a stratified medical system. They developed two models to analyze the dynamic process of patients seeking reliable hospital treatment and the strategies for achieving stratification diagnosis and appropriate referral [13]. Extend the existing framework of correlation strategies by proposing a new type of "correlation games" and identifying Nash equilibria that solve [14], examined the impact of individual vaccination choices on budget allocation and the potential for controlling vaccine-preventable diseases through the optimization of budget allocation. They achieved this by integrating evolutionary game theory with compartmental models of disease transmission [15]. The application of the game theory model based on Bayesian Nash Equilibrium (BNE) in environmentally friendly driving was studied [12]. Study utilized a doctor-patient signal game model and a two-level treatment game model to examine the impact of information spillover variables enabled by big data on the medical market, hospitals, and patients' overall satisfaction [16], assessed the influence of mobile teams on the outcomes of tuberculosis treatment in the Riyadh region of Saudi Arabia [17]. Propose a cooperative video steganography method based on game theory to achieve optimal video steganography framework design by hiding multiple images within an extended video area [18], proposed a game theory epidemiological model that combines individual preferences and government policies to analyze the impact of restrictive policies on health costs and socioeconomic issues in the COVID-19 pandemic [18], proposed a new spectrum allocation model based on learning and game theory for Internet of Things Medical Platforms (IoMT) [19], investigated the issue of seeking treatment for tuberculosis and examined how the choices made by individuals affect the occurrence of tuberculosis [20]. Proposed an epidemiological model based on evolutionary game theory for predicting infections and applied to demand forecasting of pharmaceutical inventory management problems [21], explored the impact of the interaction between seasonality, behavior, and incubation periods on public health interventions in a behavioral vaccination model aimed at minimizing the total cost of disease [4], propose a dynamic trust access control game mechanism based on blockchain to solve the problem of dynamic and trusted access control in the current open network environment [5], developed a novel model for predicting infectious diseases. They used evolutionary game theory and network structure to account for factors such as vaccine failure and incentive strategies for medical treatment. Through simulation experiments, they were able to confirm the effectiveness of their model. These studies highlight the potential of game theory in addressing complex healthcare challenges. However, existing research often focuses on individual aspects of healthcare systems, such as disease prediction or network security, with limited attention to the role of public participation in medical safety risk supervision. This gap is crucial, as public involvement can significantly enhance the effectiveness of healthcare policies and practices. but its dynamics may vary across different contexts (e.g., chronic disease management, infectious disease prevention, and drug safety). Therefore, this study aims to fill this gap by developing a tripartite evolutionary game model that incorporates the public, medical institutions, and regulatory authorities. We first analyze the theoretical interactions among these stakeholders. Then, to test the model's robustness and generalizability, we validate it using empirical data from three distinct cases: TB treatment in Saudi Arabia, COVID-19 vaccination in China, and antibiotic supervision in Vietnam. Our study provides a novel perspective on how public participation can influence the stability and effectiveness of healthcare systems, contributing to the sustainable development of intelligent medical services globally.

## 2. Problem description and hypothesis

As smart medical systems continue to develop and become more complex, medical institutions are also facing new challenges in terms of compliance and supervision of medical safety risks. On the one hand, many medical institutions may adopt some non-compliant medical practices to pursue financial interests. For example, hospitals may over-treat, operate erratically, or violate medical safety standards in an effort to increase efficiency and profitability. Government regulatory agencies, on the other hand, are responsible for formulating and enforcing medical rules and regulations to ensure safety

and order in the medical system. However, regulatory authorities also face problems such as limited regulatory resources, lagging regulatory means, and lax supervision. In order to achieve the effective implementation of smart medical safety risk supervision, the public is introduced to participate in smart medical safety risk supervision, and regulatory authorities and medical institutions conduct two-way supervision.

The article presents Fig 1, which illustrates the logical relationship between the parties involved in the three-party evolutionary game of smart medical safety risk supervision. In this model, a stable regulatory mechanism is gradually formed through interactions and games between the government, medical institutions, and the public/patients. The government formulates policies and regulations to supervise the compliance behavior of medical institutions; medical institutions may adopt different compliance strategies driven by economic interests; the public/patients influence the decisions of the government and medical institutions through feedback and participation, forming effective supervision strength.

Based on the problem descriptions and assumptions, we developed a tripartite evolutionary game model involving the public, medical institutions, and regulatory authorities. The model aims to analyze the behavioral tendencies of these stakeholders in medical safety risk supervision. The key assumptions are as follows:

**Hypothesis 1:** The subjects of the evolutionary system of intelligent healthcare safety risk regulation considering public participation behavior include healthcare institutions, regulatory authorities, and the public/patients. The participating subjects are all limited rational individuals with different interests and strategy choices that evolve over time, gradually stabilizing at the optimal strategy.

**Hypothesis 2:** It is assumed that the public/patients will expose the healthcare institutions' violations for their own safety, and the exposure rate is $\mu(0 \leq \mu \leq 1)$, indicating that the probability of exposure, affects the public's participation behavior. The probability that public $P$, Incidence of public/patient choice to actively participate in medical regulation $\delta(0 \leq \delta \leq 1)$, and the probability that it does not is $(1-\delta)$. The benefits of public participation in medical regulation are $I_{p1}$, the benefit of non-participation in regulation is $I_{p2}$, If medical institutions are allowed to engage in non-compliant behavior without strict supervision, it will lead to a decline in the quality of medical services and infrastructure, an increase in medical errors, an increase in patient safety risks, and an increase in the risk of infection. Public health risks and threats to the physical and mental health of the public/patients when violations of the law by medical institutions go unpunished $B_p$. $B_p$ is the loss caused by public inaction. The greater the loss value, the higher the participation.

**Hypothesis 3:** The probability of compliant medical care is $\eta(0 \leq \eta \leq 1)$, and the probability of non-compliant medical is $(1-\eta)$. When non-compliant medical treatment is performed, medical institutions will submit regulatory capture actions to government regulatory authorities. The revenue generated by compliant medical care is $I_{m1}$, and the input cost is $C_{m1}$. Additional income from substandard medical care is $I_{m2}$, The cost of a period of noncompliant medical care is $C_{m2}$ ($C_{m2} < C_{m1}$), The medical risks increased by non-compliant medical treatment are $V_{m2}$, the cost of colluding with the regulatory authorities is $C_c$, the fine imposed by the regulatory authorities is $\theta V_{m2}$ ($\theta$ is a penalty imposed by government regulatory authorities on medical institutions), and The reputational damage to healthcare organizations exposed to medical noncompliance is $B_m$.

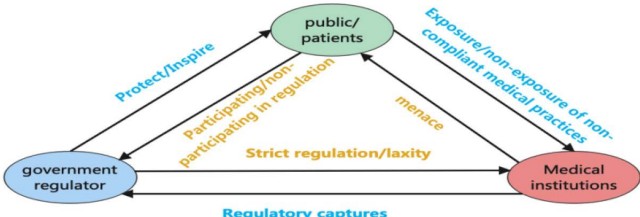

**Fig 1. The logical relationship among the subjects involved in the three-party evolutionary game.**

**Hypothesis 4:** The likelihood of rigorous oversight by government regulatory authorities is $\zeta(0 \leq \zeta \leq 1)$, and the probability of lax supervision is $(1 - \zeta)$. Under strict supervision, regulatory authorities will punish medical institutions that violate medical rules, and accept regulatory arrests when medical institutions are lax in their supervision. The advantage of strict supervision by regulatory authorities is that $I_{g1}$, the cost of supervision is $C_{g1}$, The reward benefits gained from strict supervision are $W$; When government regulation is relaxed, medical institutions conduct regulatory capture on the regulatory department, and the additional income obtained by the regulatory department is $I_{g2}$, Exposed by the public, regulatory authorities accept supervision and arrest medical institutions, and are punished by superior authorities $B_g$.

Based on these assumptions, we constructed a payoff matrix for the three stakeholders (Table 1) and derived the replication dynamic equations for each stakeholder. The stability analysis of the equilibrium points was conducted using the Jacobi matrix and Lyapunov's first method.

## 3. Model construction and analysis

Fig 2 of the flow chart shows a tripartite evolutionary game model for public health safety risk supervision, the entire process from problem description and assumptions to model verification and actual case analysis. The graphic is divided into major sections, each containing multiple steps and processes.

Based on the problem descriptions and assumptions in previous chapters, the parameters and variables involved in the model of this article were determined, and the constraints between the three themes were sorted out. This chapter gradually builds the evolutionary game model of this article from the interest payment matrix of the public, medical institutions and government regulators, and conducts replication and stability analysis of the dynamic equation system.

### 3.1. Payment matrix

Based on the above problem description and parameter settings, the interest requirements and restriction relationships between agents in the evolutionary system are further clarified. Taking safety needs as the starting point, the public/

**Table 1. Payment matrix.**

| | | Public/Patients | |
| --- | --- | --- | --- |
| | | Participate $(\delta)$ | Non-participation $(1 - \delta)$ |
| strict supervision $(\zeta)$ | Compliant medical care $(\eta)$ | $I_{g1} + W - C_{g1}$<br>$I_{m1} - C_{m1}$<br>$I_{p1}$ | $I_{g1} + W - C_{g1}$<br>$I_{m1} - C_{m1}$<br>$I_{p2}$ |
| | Non-compliant medical care $(1 - \eta)$ | $W + I_{g1} + \theta V_{m2} - C_{g1}$<br>$I_{m1} + I_{m2} - C_{m2} - \theta V_{m2} - \mu B_m$<br>$I_{p1}$ | $W + I_{g1} + \theta V_{m2} - C_{g1}$<br>$I_{m1} + I_{m2} - C_{m2} - \theta V_{m2}$<br>$I_{p2}$ |
| Lax regulation $(1 - \zeta)$ | Compliant medical care $(\eta)$ | $I_{g1} - C_{g1}$<br>$I_{m1} - C_{m1}$<br>$I_{p1}$ | $I_{g1} - C_{g1}$<br>$I_{m1} - C_{m1}$<br>$I_{p2}$ |
| | Non-compliant medical care $(1 - \eta)$ | $I_{g1} + I_{g2} - C_{g1} - \mu B_g$<br>$I_{m1} + I_{m2} - C_{m2} - \theta V_{m2} - \mu B_m$<br>$I_{p1} - (1 - \mu)B_p$ | $I_{g1} + I_{g2} - C_{g1}$<br>$I_{m1} + I_{m2} - C_{m2} - \theta V_{m2}$<br>$I_{p2} - B_p$ |

$I_{p1}$: Benefit of public participation in supervision (e.g., rewards for reporting violations, reduced personal health risks from non-compliance).

$C_{m1}$: Cost for medical institutions to comply (e.g., training staff, purchasing qualified equipment—common in healthcare supervision).

$V_{m2}$: Penalty for non-compliance: $\theta$(penalty intensity, 0-1) × $V_{m2}$ (medical risk of non-compliance, e.g., adverse reactions from expired drugs), ensuring penalties match risk severity.

$\theta V_{m2}$: Penalty amount (proportional to risk, ensuring penalties match violation severity).

$\delta$: Probability of public participation in supervision (0 = no participation, 1 = full participation).

$\zeta$: Probability of strict government supervision (0 = fully lax, 1 = fully strict).

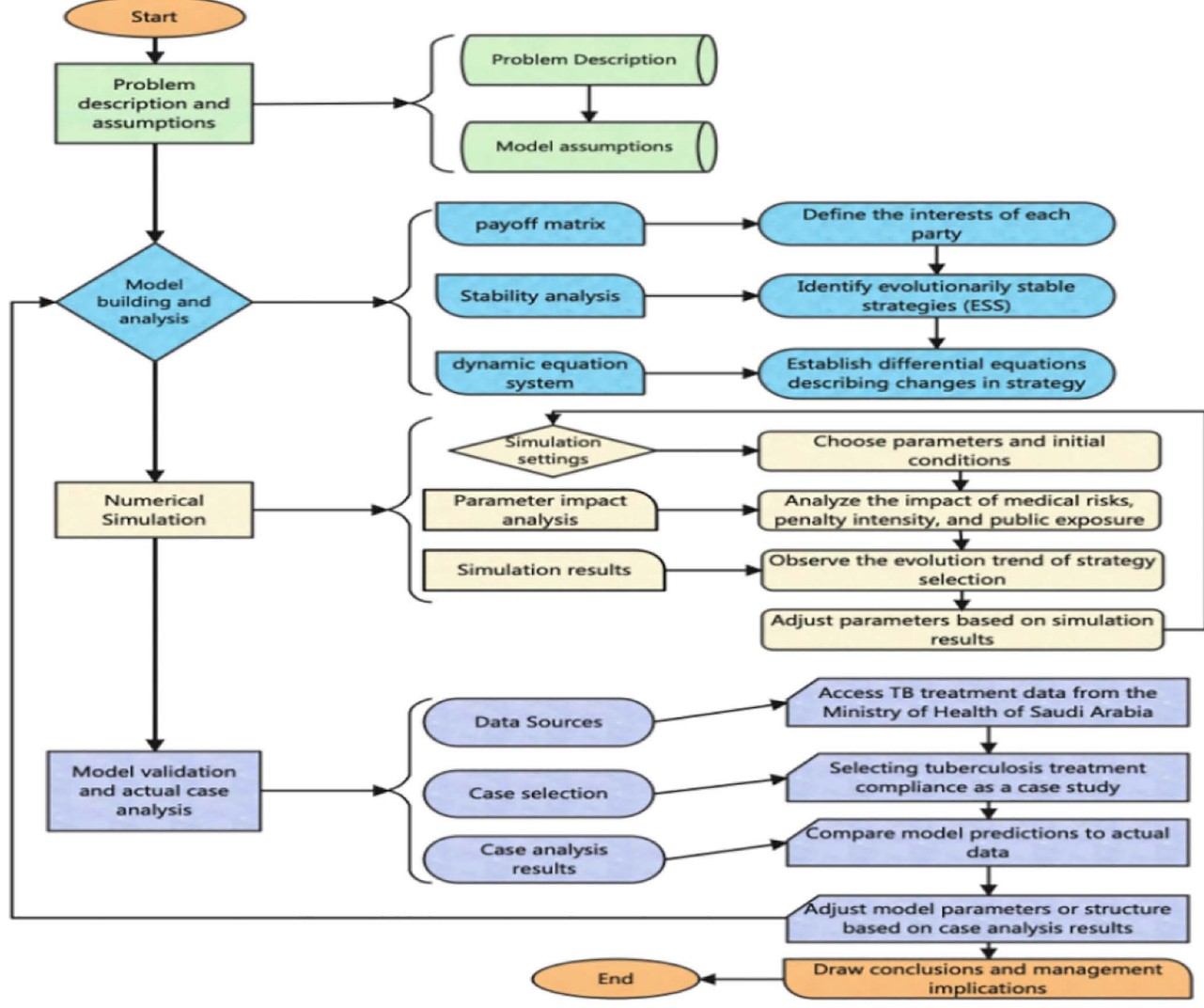

**Fig 2. Flow chart of public health safety risk supervision model based on evolutionary game theory.**

patients raise safety expectations to government regulatory authorities and medical institutions, and supervise the subject's behavior in the process. At the same time, the public's behavior is also affected by the strategic choices of medical institutions and regulatory authorities. Therefore, in order to further analyze the evolutionary game process between the three parties, a three-party game benefit matrix of the public/patients, medical institutions and government regulatory authorities was constructed, as shown in Table 1.

The expected utility functions of the three agents in the evolutionary system are as follows:

(1) The expected revenue of the public when it chooses to participate in the regulation is:

$$H_{p1} = I_{p1} - (1 - \eta)(1 - \zeta)(1 - \mu)B_p \tag{1}$$

The expected benefits when the public opts out of regulation are:

$$H_{p2} = I_{p2} - (1-\eta)(1-\zeta)B_p \tag{2}$$

The average expected benefit of the public is:

$$\overline{H}_p = \delta H_{p1} + (1-\delta)H_{p2} = \delta I_{p1} + (1-\delta)I_{p2} - (1-\mu)(1-\eta)(1-\delta\mu)B_p \tag{3}$$

The public replication dynamic equation is:

$$F_p(\delta, \eta, \zeta) = \frac{d\delta}{dt} = \delta\left(H_{p1} - \overline{H}_p\right)$$

$$= \delta\left(H_{p1} - (\delta H_{p1} + (1-\delta)H_{p2})\right)$$

$$= \delta(1-\delta)\left(H_{p1} - H_{p2}\right)$$

$$= \delta(1-\delta)\left(I_{p1} - I_{p2} + (1-\eta)(1-\zeta)\mu B_p\right) \tag{4}$$

$\frac{d\delta}{dt}$: "Speed of change in public participation rate over time" — a positive value means participation is increasing; a negative value means participation is decreasing.

$\delta(1-\delta)$: "Participation range limiter" — it ensures $\delta$ stays between 0 (0% participation) and 1 (100% participation). For example, if $\delta = 1$ (all public fully participate), $\delta(1-\delta) = 0$, so $\frac{d\delta}{dt} = 0$ (no further increase in participation).

Bracketed term: "Driving force for public participation" — e.g., $I_{p1} - I_{p2}$ (basic benefit difference between participating and not participating in supervision) and $(1-\eta)(1-\zeta)\mu B_p$ (additional benefit from successfully reporting violations when medical institutions are non-compliant $(1-\eta)$ and government supervision is lax $(1-\zeta)$, encouraging public participation).

(2) The expected benefits of compliant healthcare for healthcare institutions are:

$$H_{m1} = \delta\zeta\left(I_{m1} - C_{m1}\right) + \zeta(1-\delta)\left(I_{m1} - C_{m1}\right) + (1-\zeta)\delta\left(I_{m1} - C_{m1}\right) + (1-\zeta)(1-\delta)\left(I_{m1} - C_{m1}\right) = I_{m1} - C_{m1} \tag{5}$$

The expected benefits of noncompliant healthcare in medical institutions are:

$$H_{m2} = I_{m1} + I_{m2} - C_{m2} - (1-\zeta)C_c - \zeta\theta V_{m2} - \delta\mu B_m \tag{6}$$

The mean projected income for medical establishments is:

$$\overline{H}_m = \eta H_{m1} + (1-\eta)H_{m2}$$

$$= I_{m1} + (1-\eta)\left(I_{m2} - C_{m2} - (1-\zeta)C_c - \delta\mu B_m - \zeta\theta V_{m2}\right) + \eta C_{m1} \tag{7}$$

The replication dynamic equation governing the growth and development of medical facilities is as follows:

$$F_m(\delta, \eta, \zeta) = \frac{d\eta}{dt} = \eta\left(H_{m1} - \overline{H}_m\right)$$

$$= \eta\left(H_{m1} - (\eta H_{m1} + (1-\eta)H_{m2})\right) = \eta(1-\eta)\left(H_{m1} - H_{m2}\right)$$

$$= \eta(1-\eta)\left(-C_{m1} - I_{m2} + C_{m2} + (1-\zeta)C_c + \delta\mu B_m + \zeta\theta V_{m2}\right) \tag{8}$$

$\frac{d\eta}{dt}$: "Speed of change in medical compliance rate over time" — a positive value means compliance is increasing; a negative value means compliance is decreasing.

$\eta(1-\eta)$: "Compliance range limiter" — it ensures $\eta$ stays between 0 (0% compliance) and 1 (100% compliance). For example, if $\eta = 1$ (all institutions are compliant), $\eta(1-\eta) = 0$, so $\frac{d\eta}{dt} = 0$ (no further improvement).

Bracketed term: "Driving force for compliance" — e.g., $-C_{m1}$ (compliance cost, discouraging compliance) and $\zeta\theta V_{m2}$ (fine for non-compliance under strict supervision, encouraging compliance).

The expected benefits of strict regulation by government regulators are:

$$H_{g1} = W + I_{g1} - C_{g1} + (1-\eta)\theta V_{m2} \tag{9}$$

The expected benefits of the regulatory authorities' lax supervision are as follows:

$$H_{g2} = I_{g1} + (1-\eta)I_{g2} - C_{g1} - (1-\eta)\delta\mu B_g \tag{10}$$

The average expected revenue of the regulatory authorities is:

$$\overline{H}_g = \zeta H_{g1} + (1-\zeta)H_{g2}$$

$$= \zeta W + \zeta(1-\eta)\theta V_{m2} + (1-\eta)(1-\zeta)I_{g2} - (1-\eta)(1-\zeta)\delta\mu B_g + I_{g1} - C_{g1} \tag{11}$$

The regulatory authority's replication dynamic equation is:

$$F_g(\delta, \eta, \zeta) = \frac{d\zeta}{dt} = \zeta\left(H_{g1} - \overline{H}_g\right)$$

$$= \zeta\left(H_{g1} - (\zeta H_{g1} + (1-\zeta)H_{g2})\right) = \zeta(1-\zeta)\left(H_{g1} - H_{g2}\right)$$

$$= \zeta(1-\zeta)\left(W + (1-\eta)\theta V_{m2} + (1-\eta)\delta\mu B_g - (1-\eta)I_{g2}\right) \tag{12}$$

$\frac{d\zeta}{dt}$: "Speed of change in government supervision intensity over time" — a positive value means supervision is tightening; a negative value means supervision is relaxing.

$\zeta(1-\zeta)$: "Supervision intensity range limiter" — it ensures $\zeta$ stays between 0 (fully lax) and 1 (fully strict). For example, if $\zeta = 0$ (no supervision), $\zeta(1-\zeta) = 0$, so $\frac{d\zeta}{dt} = 0$ (no further relaxation).

Bracketed term: "Driving force for supervision adjustment" — e.g., W (benefit of strict supervision, encouraging tightening) and $-(1-\eta)I_{g2}$ (lost bribery income from lax supervision, discouraging tightening).

The replication dynamic equation of an evolutionary game is a differential equation that precisely models the temporal change in the proportion of agents adopting specific strategies. It consists of Equation (4), Equation (8), and Equation (12). The replication dynamics equation for analyzing the three-party evolutionary game in medical safety risk regulation, taking into account public involvement behavior, is as follows:

$$F_p(\delta, \eta, \zeta) = \frac{d\delta}{dt} = \delta(1-\delta)\left(I_{p1} - I_{p2} + (1-\eta)(1-\zeta)\mu B_p\right)$$
$$F_m(\delta, \eta, \zeta) = \frac{d\eta}{dt} = \eta(1-\eta)\left(-C_{m1} - I_{m2} + C_{m2} + (1-\zeta)C_c + \delta\mu B_m + \zeta\theta V_{m2}\right)$$
$$F_g(\delta, \eta, \zeta) = \frac{d\zeta}{dt} = \zeta(1-\zeta)\left(W + (1-\eta)\theta V_{m2} + (1-\eta)\delta\mu B_g - (1-\eta)I_{g2}\right) \tag{13}$$

### 3.2. Stability analysis

The first derivation of the replicated dynamic equations for the public/patients, medical institutions, and government regulators was obtained:

$$\frac{dF_p(\delta, \eta, \zeta)}{d\delta} = (1 - 2\delta)\left(I_{p1} - I_{p2} + (1 - \eta)(1 - \zeta)\mu B_p\right) \tag{14}$$

$$\frac{dF_m(\delta, \eta, \zeta)}{d\eta} = (1 - 2\eta)\left(-C_{m1} - I_{m2} + C_{m2} + (1 - \zeta)C_c + \delta\mu B_m + \zeta\theta V_{m2}\right) \tag{15}$$

$$\frac{dF_g(\delta, \eta, \zeta)}{d\zeta} = (1 - 2\zeta)\left(W + (1 - \eta)\theta V_{m2} + (1 - \eta)\delta\mu B_g - (1 - \eta)I_{g2}\right) \tag{16}$$

Set:

$$G(\zeta) = I_{p1} - I_{p2} + (1 - \eta)(1 - \zeta)\mu B_p \tag{17}$$

$$G(\delta) = -C_{m1} - I_{m2} + C_{m2} + (1 - \zeta)C_c + \delta\mu B_m + \zeta\theta V_{m2} \tag{18}$$

$$G(\eta) = W + (1 - \eta)\theta V_{m2} + (1 - \eta)\delta\mu B_g - (1 - \eta)I_{g2} \tag{19}$$

To achieve a steady state of public policy choice, $F_p(\delta, \eta, \zeta) = 0$ and $\frac{dF_p(\delta,\eta,\zeta)}{d\delta} < 0$ are required. Since $G(\zeta)$ is an increasing function, when $\zeta = 1 + \frac{I_{p1}-I_{p2}}{(1-\eta)\mu B_p}$, there is $G(\zeta) = 0$ and $\frac{dF_p(\delta,\eta,\zeta)}{d\delta} \equiv 0$, the public cannot be sure of its stability strategy at this time. When $\left.\frac{dF_p(\delta,\eta,\zeta)}{d\delta}\right|_{\delta=0} < 0$, $G(\zeta) < 0$, that is, $\zeta > 1 + \frac{I_{p1}-I_{p2}}{(1-\eta)\mu B_p}$, $\delta = 0$ is the evolutionary stable point, and the public's choice not to participate in supervision is its stable strategy. When $\left.\frac{dF_p(\delta,\eta,\zeta)}{d\delta}\right|_{\delta=1} < 0$, $G(\zeta) > 0$, that is, $\zeta < 1 + \frac{I_{p1}-I_{p2}}{(1-\eta)\mu B_p}$, $\delta = 1$ is the evolutionary stable point, and the public chooses to participate in supervision as its stability strategy. The phase diagram of public policy evolution is shown in Fig 3(a).

The $\delta$ corresponding to all points of the section in the phase diagram (Fig 3(a)) is the stability strategy of the public, and the $\delta$ on the upper side of the section tends to be $\delta = 1$, The public's stabilizing strategy tends to involve participation in supervision. When the government relaxes supervision, medical institutions do not comply with medical regulations, and medical institutions are not punished for violating medical regulations, the losses from medical risks and threats to the public's physical and mental health will be reduced, and the public's stabilization strategy will shift to unsupervised, that is, the lower part of the cross section, the evolutionary stable point tends to $\delta = 0$.

Policy options to achieve steady-state medical institutions, $F_m(\delta, \eta, \zeta) = 0$ and $\frac{dF_m(\delta,\eta,\zeta)}{d\eta} < 0$ are required. Since $G(\delta)$ is a decreasing function, when $\delta = 1 + \frac{C_{m1}+I_{m2}-C_{m2}-(1-\zeta)C_c-\zeta\theta V_{m2}}{\mu B_m}$, $G(\delta) = 0$, $\frac{dF_m(\delta,\eta,\zeta)}{d\eta} \equiv 0$, Healthcare organizations then cannot determine their stability strategy, When $\left.\frac{dF_m(\delta,\eta,\zeta)}{d\eta}\right|_{\eta=1} < 0$, there is $G(\delta) > 0$, that is $\delta > 1 + \frac{C_{m1}+I_{m2}-C_{m2}-(1-\zeta)C_c-\zeta\theta V_{m2}}{\mu B_m}$, $\eta = 1$ is the point of evolutionary equilibrium. Compliant medical care is a stable strategy for medical institutions. When $\left.\frac{dF_m(\delta,\eta,\zeta)}{d\eta}\right|_{\eta=0} < 0$, $G(\delta) < 0$, that is, $\delta < 1 + \frac{C_{m1}+I_{m2}-C_{m2}-(1-\zeta)C_c-\zeta\theta V_{m2}}{\mu B_m}$, $\eta = 0$ is the evolutionary stability point, Medical institutions choose non-compliant care as their stabilization strategy. The strategic evolution stage diagram of building medical institutions is shown in Fig 3(b).

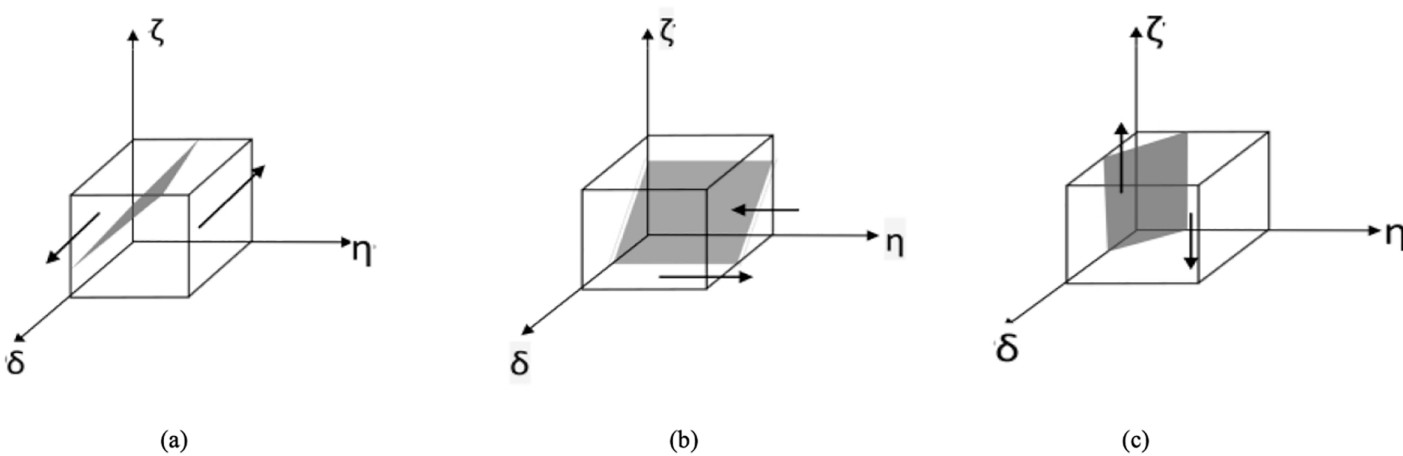

**Fig 3. Phase diagram of tripartite agent strategy evolution.**

In the phase diagram (Fig 3(b)), the $\eta$ Corresponding to all the points in this section is the Stability Strategy for Healthcare Organizations, and the $\eta$ at the lower side of the section tends to be $\eta = 0$, The stabilization strategy of medical institutions tends to choose non-compliant medical treatment. If the additional income of medical institutions from non-compliant medical treatment decreases, the government strictly regulates and intensifies penalties, and the public participates in supervision, the probability of exposure of non-compliant medical treatment increases, and the strategic choice of medical institutions will shift to compliant medical care, that is, the upper part of the cross-section, and the evolutionary stability point tends to be $\eta = 1$.

To achieve the stable state of the regulator's policy selection, $F_g(\delta, \eta, \zeta) = 0$ and $\frac{dF_g(\delta,\eta,\zeta)}{d\zeta} < 0$ are required. Since $G(\eta)$ is a decreasing function, when $\eta = 1 + \frac{W}{\theta V_{m2} + \delta \mu B_g - I_{g2}}$, $G(\eta) = 0$, $\frac{dF_g(\delta,\eta,\zeta)}{d\zeta} \equiv 0$, then the regulator can not determine its stability strategy. When $\left. \frac{dF_g(\delta,\eta,\zeta)}{d\zeta} \right|_{\zeta=1} < 0$, $G(\eta) > 0$, that is, $\eta < 1 + \frac{W}{\theta V_{m2} + \delta \mu B_g - I_{g2}}$, $\zeta = 1$ is the evolutionary stability point, and the regulatory authorities choose strict supervision as the evolutionary stability strategy. When $\left. \frac{dF_g(\delta,\eta,\zeta)}{d\zeta} \right|_{\zeta=1} < 0$, $G(\eta) < 0$, that is, $\eta > 1 + \frac{W}{\theta V_{m2} + \delta \mu B_g - I_{g2}}$, $\zeta = 0$ is the evolutionary stability point, and the regulatory department chooses loose supervision as its stability strategy. The phase diagram of the strategy evolution of the constructed regulatory department is shown in Fig 3(c).

In the phase diagram (Fig 3(c)), the $\zeta$ All points on the cross section correspond to the stable strategy of the medical institution, and the $\zeta$ on the upper side of the cross section tends to be $\zeta = 0$, and The stabilization strategy of government regulatory authorities tends to choose loose supervision. If the additional revenue brought by the supervision and arrest of medical institutions to the government supervision department is reduced, the public exposure supervision department will accept the supervision and arrest of medical institutions, thereby receiving more penalties from superior departments. The government will punish non-compliance under strict supervision. As medical institutions impose more penalties, government regulatory authorities will shift their strategic choices to strict supervision, that is, in the lower part of the cross-section, the evolutionary stability point tends to $\zeta = 1$.

The stability strategies for each participant can be visualized through phase diagrams. As shown in Fig 3, these diagrams illustrate how the choices of one party are influenced by the strategies of the others. For the public, the choice to participate in supervision ($\delta = 1$) becomes the stable strategy when the government's supervision is relatively lax, as this increases the potential benefit of reporting non-compliance. For medical institutions, compliant medical care ($\eta = 1$) becomes the stable strategy when public participation is high and government penalties are stringent, making the cost of

non-compliance (reputational damage and fines) outweigh the potential gains. For government regulators, strict supervision ($\zeta = 1$) is the stable strategy when the probability of non-compliant medical care is high, as the rewards for enforcement and penalties from superiors for laxity become more significant.

### 3.3. Analysis of the stability of the equilibrium point in a tripartite evolutionary game system

Based on the principle of maximizing each player's own interests, the players of evolutionary game adjust their own strategies according to the strategies of other players, and finally make the strategies of participating players tend to be stable, The stable strategy in an evolutionary game is referred to as the Evolutionarily Stable Strategy (ESS). ESS is the subset of equilibrium points of the corresponding replicating dynamic equation set. In order to obtain the equilibrium point when Public/patients, medical institutions, government regulatory authorities all reach the stable state of the evolutionary game, so that $F_p(\delta, \eta, \zeta) = 0$, $F_m(\delta, \eta, \zeta) = 0$, $F_g(\delta, \eta, \zeta) = 0$, to obtain the eight equilibrium points of the replicated dynamic equations, $E_1(0, 0, 0)$, $E_2(1, 0, 0)$, $E_3(0, 1, 0)$, $E_4(0, 0, 1)$, $E_5(1, 1, 0)$, $E_6(1, 0, 1)$, $E_7(0, 1, 1)$, $E_8(1, 1, 1)$.

The Jacobi matrix corresponding to the tripartite evolutionary game system is:

$$J = \begin{bmatrix} \frac{\partial F_p(\delta,\eta,\zeta)}{\partial \delta} & \frac{\partial F_p(\delta,\eta,\zeta)}{\partial \eta} & \frac{\partial F_p(\delta,\eta,\zeta)}{\partial \zeta} \\ \frac{\partial F_m(\delta,\eta,\zeta)}{\partial \delta} & \frac{\partial F_m(\delta,\eta,\zeta)}{\partial \eta} & \frac{\partial F_m(\delta,\eta,\zeta)}{\partial \zeta} \\ \frac{\partial F_g(\delta,\eta,\zeta)}{\partial \delta} & \frac{\partial F_g(\delta,\eta,\zeta)}{\partial \eta} & \frac{\partial F_g(\delta,\eta,\zeta)}{\partial \zeta} \end{bmatrix}$$

$$= \begin{bmatrix} (1-2\delta)(I_{p1}-I_{p2}+(1-\eta)(1-\zeta)\,\mu B_p) & -\delta\,(\delta-1)(\zeta-1)\,\mu B_p & -\delta\,(\delta-1)(\eta-1)\,\mu B_p \\ \mu\eta\,(1-\eta)\,B_m & (1-2\eta)(-C_{m1}-I_{m2}+C_{m2}+(1-\zeta)\,C_c+\delta\mu B_m+\zeta\theta V_{m2}) & \eta\,(1-\eta)(\theta V_{m2}-C_c) \\ \delta\,(1-\delta)(1-\eta)\,\mu B_g & \zeta\,(\zeta-1)(\theta V_{m2}+\delta\mu B_m-I_{g2}) & (1-2\zeta)(W+(1-\eta)\,\theta V_{m2}+(1-\eta)\,\delta\mu B_g-(1-\eta)\,I_{g2}) \end{bmatrix}$$

Simplified Stability Analogy for Jacobian Matrix:

Think of the system as a "bowl with a small ball":

- Negative eigenvalues (the core condition for stability) mean that pushing the ball slightly (a small shock, e.g., temporary lax supervision) will make it roll back to the bottom of the bowl (stable state).

- Positive eigenvalues would mean the ball rolls out of the bowl (system collapse, e.g., public participation keeps dropping until no one joins).

This analogy confirms the system will not oscillate endlessly — even with small disruptions, it will eventually stabilize (critical for reliable policy recommendations).

According to Lyapunov's first technique, if all the eigenvalues of the Jacobi matrix have non-positive real components, then the equilibrium point is asymptotically stable. Thus, the assessment of the asymptotic stability of the equilibrium point can be determined based on the eigenvalues of the Jacobi matrix. If all eigenvalues possess negative real components, the equilibrium point corresponds to the evolutionary equilibrium point of the evolutionary game. The stability of every equilibrium point is examined, as indicated in Table 2.

**Proposition 1:** The system has and has only one evolutionary stable point $E_6(1, 0, 1)$ when $C_{m2} - C_{m1} - I_{m2} + \theta V_{m2} + \mu B_m < 0$ and $I_{p2} - I_{p1} < 0$.

Proof: The condition for the system to have an evolutionary stable point is that all eigenvalues are negative. This is demonstrated in Table 2, where the eigenvalues satisfy the given conditions. Consequently, the system has only one evolutionary stable point and an evolutionary stable strategy. It is evident that Proposition 1 is valid, and the proof is now complete.

**Proposition 2:** When $C_{m1} - C_{m2} + I_{m2} - C_c - \mu B_m < 0$ and $I_{p2} - I_{p1} < 0$, the system has and has only one evolutionary stable point $E_8(1, 1, 1)$.

**Table 2. Stability analysis of equilibrium points.**

| Equalization point | Matrix eigenvalue | | | | Stability | Conditions |
|---|---|---|---|---|---|---|
| | $\lambda_1$ | $\lambda_2$ | $\lambda_3$ | Real part symbol | | |
| $E_1(0,0,0)$ | $I_{p1}-I_{p2}+\mu B_p$ | $C_{m2}-C_{m1}-I_{m2}+C_c$ | $W-I_{g2}+\theta V_{m2}$ | $(+,-,+)$ | Unstable point | |
| $E_2(1,0,0)$ | $I_{p2}-I_{p1}-\mu B_p$ | $\boldsymbol{C_{m2}-C_{m1}-I_{m2}+C_c+\mu B_m}$ | $W-I_{g2}+\theta V_{m2}+\mu B_g$ | $(-,-,+)$ | Unstable point | |
| $E_3(0,1,0)$ | $I_{p1}-I_{p2}$ | $C_{m1}-C_{m2}+I_{m2}-C_c$ | $W$ | $(+,-,+)$ | Unstable point | |
| $E_4(0,0,1)$ | $I_{p1}-I_{p2}$ | $C_{m2}-C_{m1}-I_{m2}+\theta V_{m2}$ | $I_{g2}-W-\theta V_{m2}$ | $(+,+,-)$ | Unstable point | |
| $E_5(1,1,0)$ | $I_{p2}-I_{p1}$ | $\boldsymbol{C_{m1}-C_{m2}+I_{m2}-C_c-\mu B_m}$ | $W$ | $(-,-,+)$ | Unstable point | |
| $E_6(1,0,1)$ | $I_{p2}-I_{p1}$ | $C_{m2}-C_{m1}-I_{m2}+\theta V_{m2}+\mu B_m$ | $I_{g2}-W-\theta V_{m2}-\mu B_g$ | $(-,-,-)$ | ESS | ① |
| $E_7(0,1,1)$ | $I_{p1}-I_{p2}$ | $C_{m2}-C_{m1}+I_{m2}-\theta V_{m2}$ | $-W$ | $(+,+,-)$ | Unstable point | |
| $E_8(1,1,1)$ | $I_{p2}-I_{p1}$ | $\boldsymbol{C_{m1}-C_{m2}+I_{m2}-C_c-\mu B_m}$ | $-W$ | $(-,-,-)$ | ESS | ② |

Note: Indicates that the symbol is uncertain and unknown,

① $C_{m2}-C_{m1}-I_{m2}+\theta V_{m2}+\mu B_m < 0$, $I_{p2}-I_{p1} < 0$

② $C_{m1}-C_{m2}+I_{m2}-C_c-\mu B_m < 0$, $I_{p2}-I_{p1} < 0$

Proof: The condition for the system to have an evolutionary stable point is that all eigenvalues must be negative. Table 2 shows that the eigenvalues satisfy this condition, indicating that the system has only one evolutionary stable point. Therefore, the system has an evolutionary stable strategy. It is evident that proposition 2 is valid, and the proof is complete.

## 4. Comprehensive simulation and case analysis

### 4.1. Numerical simulation analysis based on XGBoost and SHAP

A simulation analysis was undertaken to validate the effectiveness of the stability analysis and assess the impact of key factors on the behavior evolution of the system. The investigation focused on the stable equilibrium point of the system. According to the actual situation, in medical safety risk supervision, since the units of parameters such as the income of each entity and medical risks are inconsistent, a unified scale process is performed so that the unit of each parameter is 1.

To demonstrate the internal dynamics and behavioral evolution predicted by our tripartite model, we first conduct a numerical simulation using the parameters from our primary case study—tuberculosis (TB) treatment in Saudi Arabia—as a baseline scenario. This allows us to systematically explore the impact of key factors on the system's evolution and stability under a controlled setting. Subsequently, in Section 4.3, we will validate the model's broader applicability by testing its predictions against real-world data from all three diverse healthcare cases.

[22] examined the matter of treatment adherence among tuberculosis (TB) patients in Saudi Arabia, with a particular focus on the Jeddah region. They compared directly observed treatment short-course (DOTS) supervised by community mobile outreach teams with facility-based DOTS treatment in TB patients on non-treatment rates and perceptions of treatment. The numerical simulation data in this article refer to literature [22].

The return $I_{p1}$ of public participation in security risk supervision is 30, the return $I_{p2}$ of non-participation in supervision is 20, the probability $\mu$ of exposing noncomplianfit behavior is 0.8, and the loss $B_p$ caused by inaction is 15. The benefit $I_{m1}$ and cost $C_{m1}$ of Compliant medical care are 40 and 30 respectively. The additional revenue $I_{m2}$ obtained by non-compliant medical care is 20, the cost $C_{m2}$ is 15, the penalty for non-compliance $\theta$ is 0.6, The medical risks increased by non-compliant medical treatment are $V_{m2}$ is 20, the loss $B_m$ due to exposure is 70, and the cost $C_c$ of capturing the regulatory authorities is 15. The income $I_{g1}$ of strict supervision by the supervisory department is 30, The reward benefits gained from strict supervision are W is 25, the regulatory cost $C_{g1}$ is 15, the additional benefit $I_{g2}$ obtained by the regulatory capture is

20, and the penalty $B_g$ by the superior department if it is found to accept the regulatory capture is 25. The specific values of each parameter are as follows:

$$I_{p1} = 30, \ I_{p2} = 20, \ \mu = 0.08, \ B_p = 15, \ I_{m1} = 40,$$

$$C_{m1} = 30, \ I_{m2} = 20, \ C_{m2} = 15$$

$$\theta = 0.6, \ V_{m2} = 20, \ B_m = 70, \ C_c = 15, \ I_{g1} = 30,$$

$$W = 25, \ C_{g1} = 15, \ I_{g2} = 20, \ B_g = 25$$

According to the replicated dynamic equations and parameter assignments, as shown in Fig 4, if the replicated dynamic equations evolve 200 times over time, no matter what the initial strategy probability of the game player is, the system will eventually converge to $E_8(1, 1, 1)$, which is consistent with the evolutionary stability analysis results and verifies the effectiveness of the model.

As indicated by Fig 4, we conducted numerical simulations to validate the effectiveness of our model and to assess the impact of key factors on system behavior. The results indicated that medical risks, penalty intensity, and exposure rate significantly influenced the stability and effectiveness of the healthcare system. For instance, higher medical risks (Vm2) and exposure rates (μ) resulted in increased public participation and medical compliance. Likewise, higher penalty intensity (θ) effectively deterred non-compliant medical practices.

Fig 5 illustrates the evolution of the public participation rate (δ), medical compliance rate (η), and government supervision rate (ζ) over normalized time under simulated data conditions. The system rapidly attains a stable state within a brief period.

The public participation rate, medical compliance rate, and government supervision rate all approach 1.0, suggesting that the system is stable and effective, with high levels of public participation, medical compliance, and government oversight.

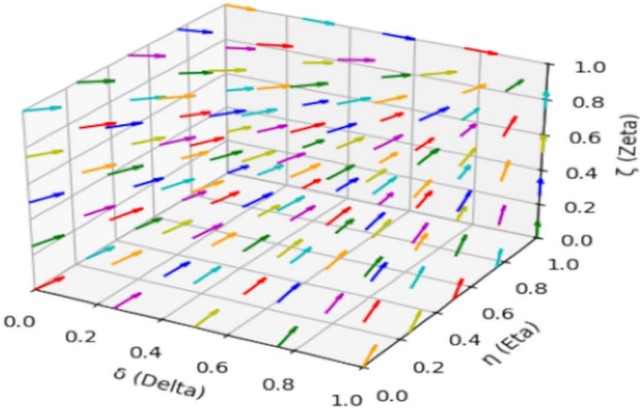

**Fig 4. Stability test of equilibrium point $E_8(1, 1, 1)$.**

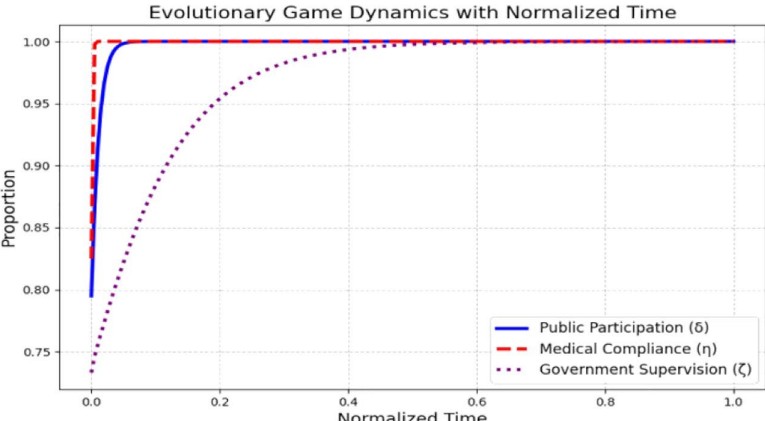

**Fig 5. Evolutionary game dynamics with normalized time.**

The vertical axis in Fig 6 lists the feature names, arranged by their importance from top to bottom. The length of each bar represents the feature's impact on the model, with longer bars indicating greater significance. This graphical representation reveals that key features like medical risk, penalty intensity, and exposure rate not only dominate but also exhibit varying effects across different SHAP value spans. For example, the medical risk feature shows a wide range of SHAP values, indicating its influence fluctuates significantly across various scenarios. Conversely, penalty intensity tends to have

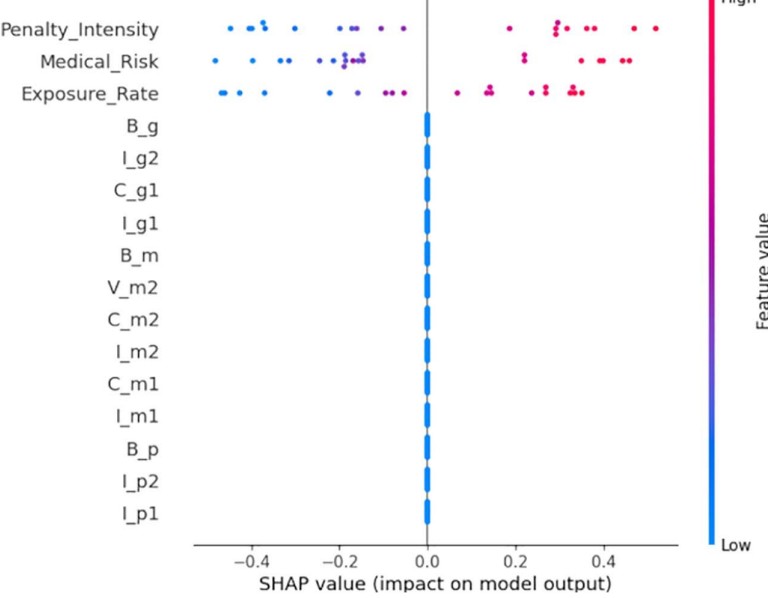

**Fig 6. SHAP Summary Plot of Key Parameters (Medical Risk Vm2), Penalty Intensity θ), Exposure Rate μ) on Medical Compliance η).** Note: Fig 6 shows that medical risk (Vm2) is the most influential parameter on medical compliance (η) — its SHAP value range (−1.2 to 0.8) is wider than penalty intensity (θ: −0.6 to 0.4) and exposure rate (μ: −0.5 to 0.3). High Vm2 (red dots) correlates with positive SHAP values, meaning higher medical risks (e.g., antibiotic resistance) drive medical institutions to adopt compliant practices. This aligns with the study's finding that mitigating medical risks is a priority for improving compliance. The X-axis represents the SHAP value (impact on η), the Y-axis lists parameters ranked by importance, and the color bar indicates parameter values (low = blue, high = red).

a more uniform impact, as evidenced by its relatively shorter yet densely colored bar. By examining the color gradients within each bar, we can discern whether a feature's effect leans towards increasing or decreasing the model's prediction values. These nuanced insights from the SHAP analysis are crucial for comprehending the model's dynamics and for guiding targeted adjustments or interventions based on these pivotal factors.

Fig 7 illustrates the importance hierarchy of features, with the horizontal axis depicting the average absolute SHAP value of each feature (indicating its average impact on model output). Features are arranged in order of descending importance. The figure reveals that medical risk is the most significant feature for model predictions, succeeded by penalty intensity and exposure rate. This indicates that medical risk is pivotal in model predictions, and that both penalty intensity and exposure rate also exert a substantial influence on the system's evolution.

As depicted in Fig 8, the system's stability is depicted as a function of three primary parameters. By manipulating these parameters, the optimal combination can be identified to attain the highest level of system stability. Although varying $Vm_2$ values do influence absolute stability, they have minimal effect on the trend of stability changes. With all other variables remaining constant, altering the medical risk value does not impact the stability of the system's evolution. The evolutionary stable point persists at $E_8(1,1,1)$. An increase in medical risks will hasten the transformation of medical institutions towards more reliable and compliant care. Concurrently, the likelihood of public involvement in medical risk regulation also increases. Consequently, as medical risks escalate, medical institutions and the public are likely to intensify their responsibilities and actively engage in medical safety risk supervision, thereby mitigating medical safety risks.

Fig 9 illustrates the impact of Penalty Intensity on model output, factoring in Medical Risk. The horizontal axis denotes the value of Penalty Intensity, while the vertical axis corresponds to the respective SHAP value. The color coding indicates the level of Medical Risk, ranging from low (blue) to high (red).

The graph depicts a positive correlation between Penalty Intensity and model output. In other words, as Penalty Intensity rises, so does the model output value. High Medical Risk (red points) amplifies the positive effect of Penalty Intensity

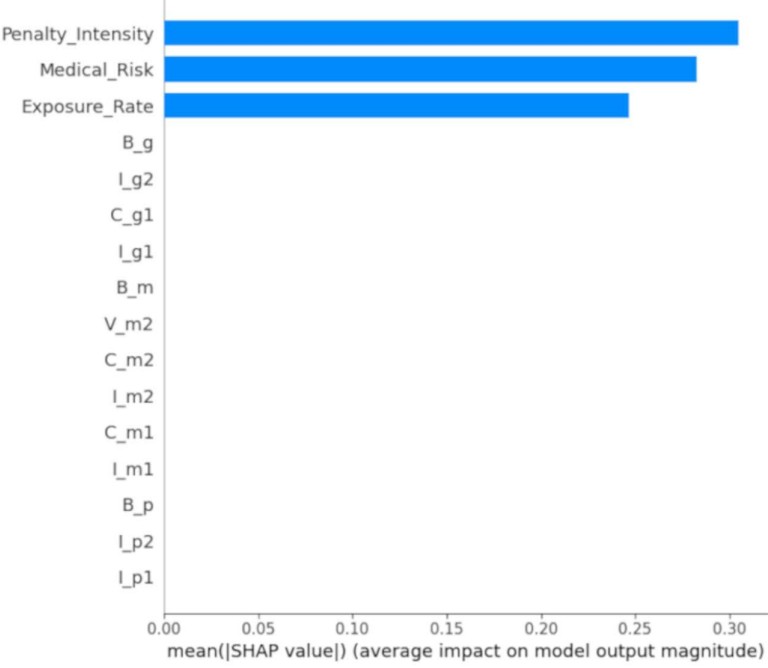

**Fig 7. SHAP feature importance ranking.**

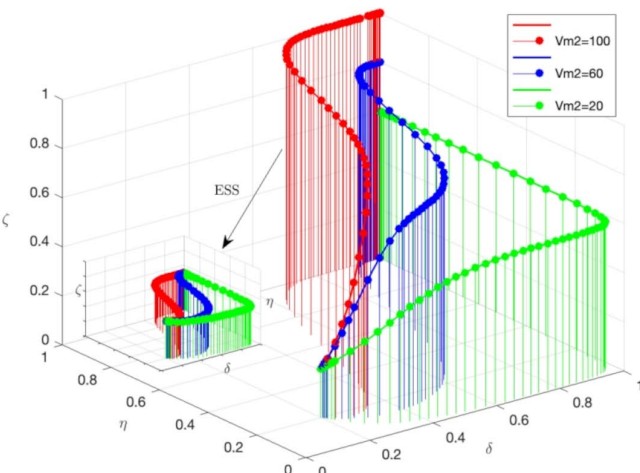

**Fig 8. 3D Surface plots of system stability with different Vm$_2$ Values.** Note: The simulation is based on the parameter values from the Saudi TB case as a baseline scenario.

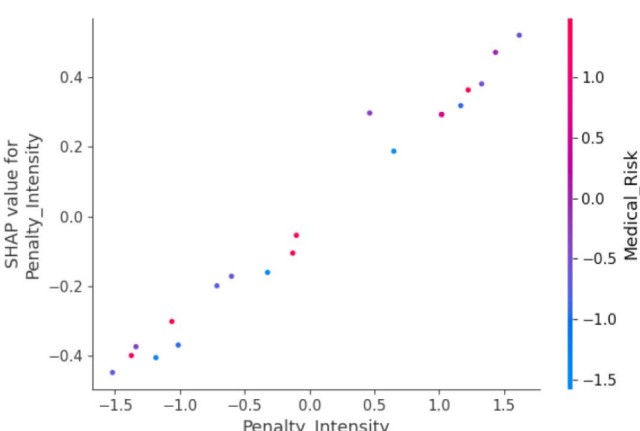

**Fig 9. Dependence of penalty intensity on model output.**

on model output. This suggests that increasing the severity of penalties and prioritizing medical risks are key strategies to enhance compliance behavior.

Fig 10 illustrates the relationship between system stability and three key parameters. By fine-tuning these parameters, the most effective combination can be determined to maximize system stability. Variations in θ values have a profound effect on the system's stability. When other parameters are held constant, changing θ impacts the stability of the system's progression. An increase in θ from 0.2 to 0.8 leads to a significant improvement in system stability. Notably, at a θ value of 0.8, the system's stability is at its highest, with the corresponding curve peaking on the graph. The graph suggests that higher θ values correlate with a greater inclination for medical institutions and the public to enhance their roles and actively engage in medical safety risk oversight, thereby lowering medical safety risks. In contrast, lower θ values are associated with reduced system stability and a curve positioned lower on the graph. Irrespective of θ value changes, the evolutionary stable state (ESS) remains at (1,1,1). An elevation in θ values accelerates the transformation rate of medical

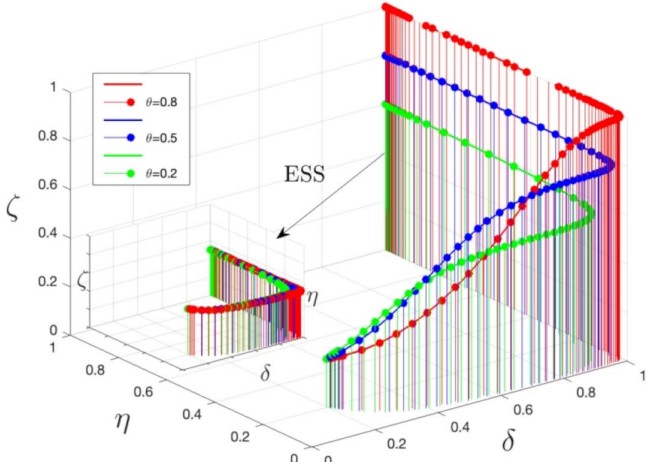

**Fig 10. 3D surface plots of system stability with different θ values.** Note: The simulation is based on the parameter values from the Saudi TB case as a baseline scenario.

institutions and increases the probability of public involvement in medical risk oversight. Consequently, as θ values rise, both medical institutions and the public are more apt to amplify their responsibilities and actively participate in medical safety risk oversight, which in turn reduces medical safety risks.

As illustrated in Fig 11, as the severity of punishment increases, so does the regulatory authorities' strategy of opting for stricter regulation.

However, it is important to note that the effectiveness of such measures is not solely dependent on the harshness of the penalties. The regulatory framework must also be designed to be fair and transparent, ensuring that the rules are applied consistently across all sectors. This approach not only deters potential offenders but also builds public trust in the regulatory system.

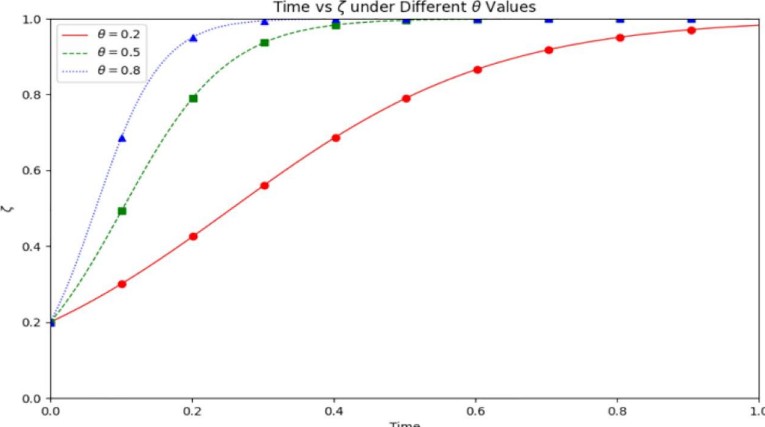

**Fig 11. Influence of penalty intensity on regulatory authorities' evolutionary strategies.** Note: The simulation is based on the parameter values from the Saudi TB case as a baseline scenario.

As depicted in Fig 12, the study examines the effects of changes in public participation and exposure rates on safety and medical risk regulation, as well as on the evolution and stability of system behavior. By adjusting the exposure rate, it becomes evident that with a low exposure rate, the system's equilibrium point gradually shifts towards $E_6(1, 0, 1)$. Conversely, as the exposure rate increases, the stable point of the system's evolution moves towards $E_8(1, 1, 1)$. The results suggest that as public oversight increases and exposure rates rise, there is a corresponding increase in the likelihood of reputation damage due to the revelation of non-compliant medical practices. Moreover, the reputation loss from exposure surpasses any additional benefits gained from non-compliant medical treatments. Consequently, medical institutions opt for compliant medical care. The strategic choices of the public, medical institutions, and government regulatory authorities stabilize in a combination of participation, compliant medical care, and stringent oversight. In the face of public scrutiny and high exposure, medical institutions and government regulatory bodies are inclined to fulfill their roles within the system, ensuring compliance in the medical system, enforcing strict supervision, and enhancing the effectiveness of medical risk safety oversight.

As illustrated by the left graph in Fig 13, the medical compliance rate peaks under conditions of high public participation and low medical risks. This indicates that enhancing public participation and mitigating medical risks can effectively improve the medical compliance rate. The right side of Fig 13 clearly shows that the government supervision rate approaches 1 when public participation is high and medical risks are low. This suggests that high public participation and low medical risks contribute to efficient government supervision. In conclusion, both the public participation rate and medical risks are significant factors that influence the stability and effectiveness of the system. By proactively increasing public participation and reducing medical risks, the compliance and supervision outcomes of the medical system can be markedly enhanced, leading to improved public health safety management.

As depicted in Fig 14, the graph illustrates system stability as a function of three primary parameters. By fine-tuning these parameters, the optimal combination can be identified to attain the highest level of system stability. Different μ values (public exposure) have a substantial effect on the stability of the system. When the μ value rises from 0.5 to 0.9, the system's stability is notably enhanced. Particularly, when the μ value is 0.9, the system's stability is at its peak, and the corresponding curve is situated at the graph's highest point. The figure indicates that higher μ values lead to increased public attention to medical safety issues, prompting medical institutions and the public to more actively fulfill their responsibilities and engage in medical safety risk supervision, thus decreasing medical safety risks. Conversely, when the μ value is low, the system's stability diminishes, and the corresponding curve is positioned lower on the graph. Regardless of the

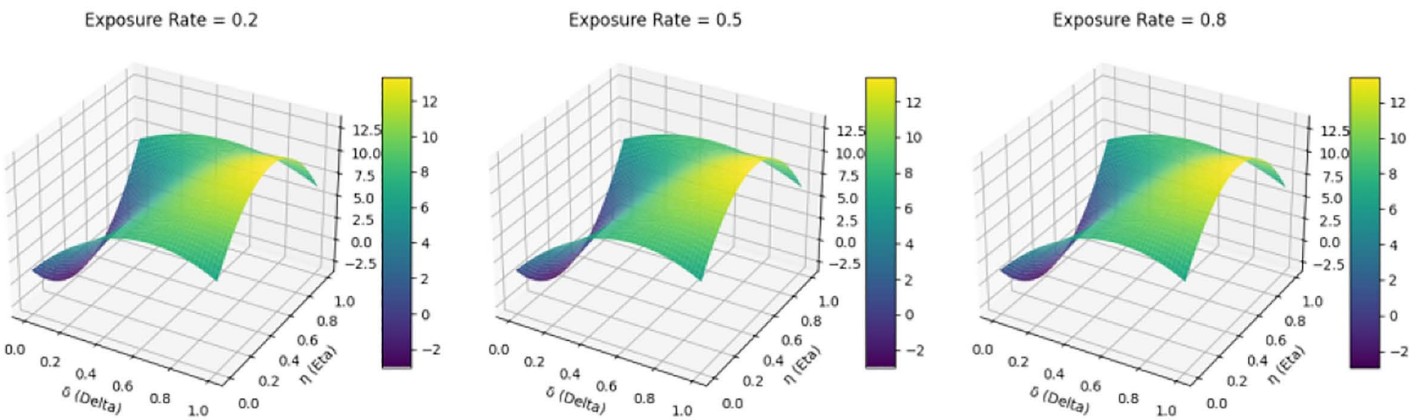

**Fig 12. Evolution trend of the system when exposure rate changes.** Note: The simulation is based on the parameter values from the Saudi TB case as a baseline scenario.

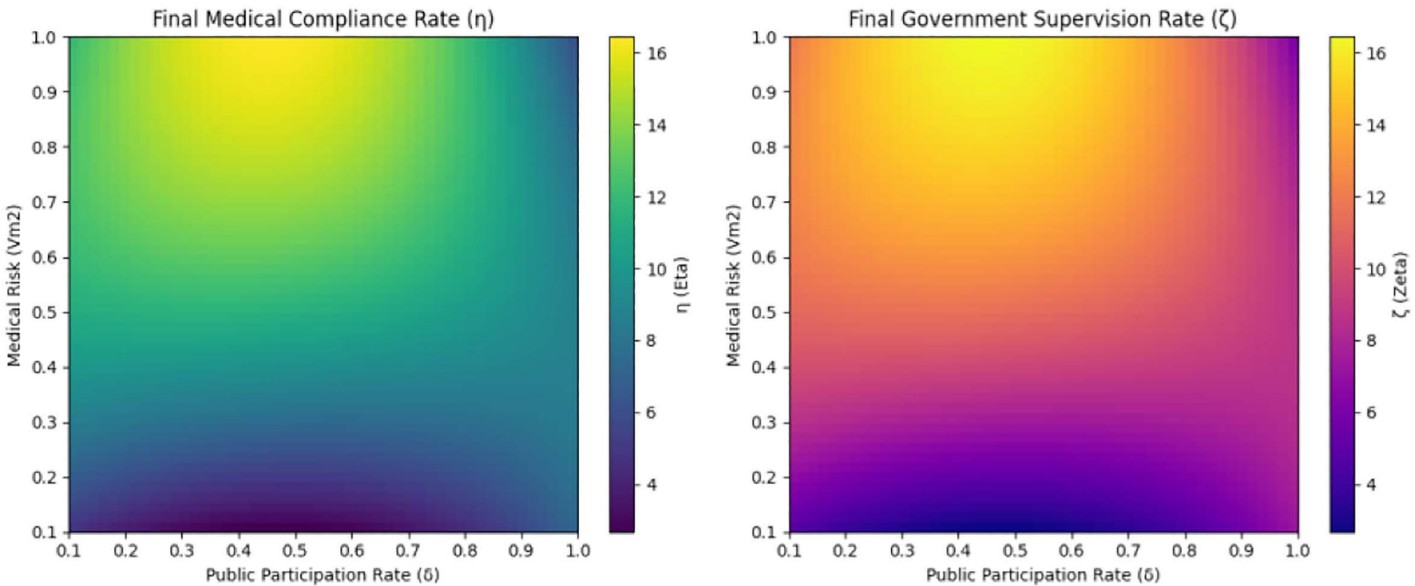

**Fig 13. Final Medical Compliance Rate (η) and Government Supervision Rate (ζ) Across Different Public Participation Rates (δ) and Medical Risks (Vm2).**

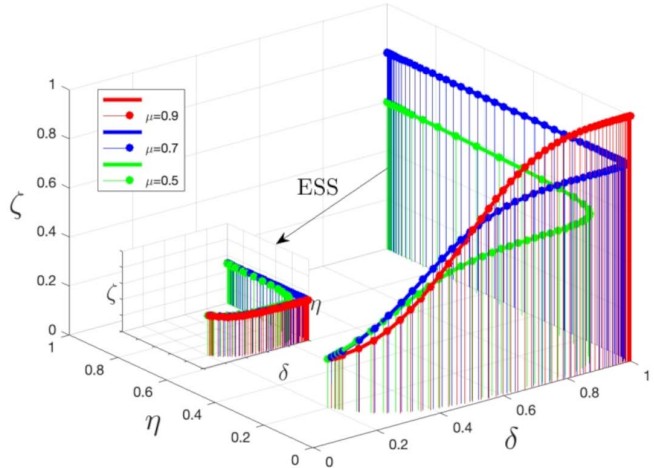

**Fig 14. 3D surface plots of system stability with different μ values.** Note: The simulation is based on the parameter values from the Saudi TB case as a baseline scenario.

μ value's fluctuations, the evolutionary stable point (ESS) persists at (1,1,1). As the μ value ascends, public exposure to medical safety issues intensifies, accelerating the transformation of medical institutions, and correspondingly, the likelihood of public involvement in medical risk supervision increases. Consequently, with an increase in the μ value, medical institutions and the public are more inclined to reinforce their responsibilities and actively engage in medical safety risk supervision, thereby mitigating medical safety risks.

The strategic decisions made by medical institutions are influenced by the public's exposure to substandard medical care, as illustrated in Fig 15. With an increase in exposure, the convergence rate of medical institutions in selecting

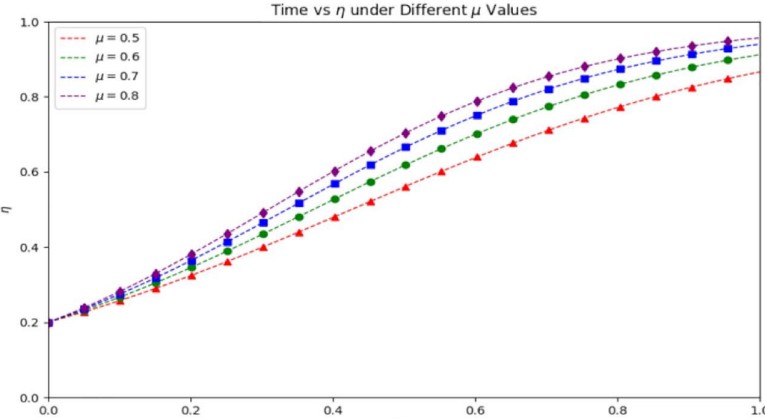

**Fig 15. The impact of exposure rate on the evolution strategy of medical institutions.** Note: The simulation is based on the parameter values from the Saudi TB case as a baseline scenario.

compliant medical treatments gradually rises. When the exposure rate is below a certain threshold, medical institutions may adopt a Lucky mentality (Lucky mentality is a Chinese phrase that translates to "Take a chance" in English, which means a mentality of taking risks or hoping for luck), and the likelihood of choosing non-compliant medical treatments increases, eventually stabilizing and opting for non-compliant medical strategies. When the exposure probability exceeds 0.7, medical institutions are more inclined to select compliant medical strategies.

As depicted in Fig 16, with an increase in the exposure rate, the regulator's strategic choice stabilizes at strict supervision. As the exposure rate continues to rise, the convergence speed of government regulators' choice towards strict supervision accelerates. The regulator's strategic choice remains unaffected by exposure changes and is influenced solely by the convergence rate of its strategy selection.

### 4.2. Sensitivity analysis

Sensitivity analysis revealed that medical risks, exposure rates, and penalty intensity were critical factors affecting the system's stability. Increasing these factors generally improved the stability and effectiveness of the healthcare system.

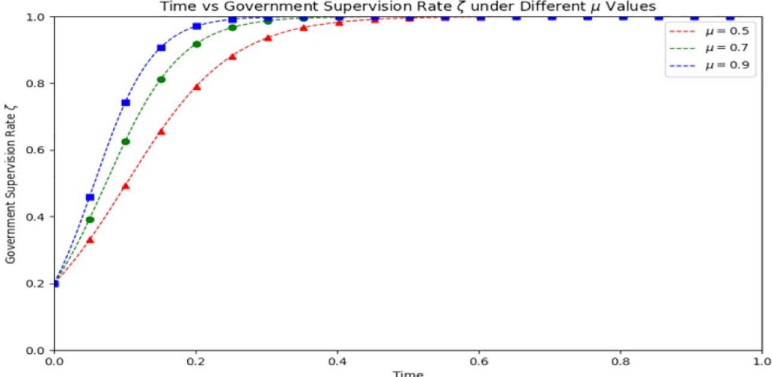

**Fig 16. The impact of exposure on the evolution strategy of government regulatory agencies.** Note: The simulation is based on the parameter values from the Saudi TB case as a baseline scenario.

Previous numerical simulations have revealed that three factors—exposure rate, medical risk, and government penalty levels—each have varying impacts on public participation, medical institutions, and government supervision. Consequently, conducting sensitivity analyses on these factors can offer a deeper understanding of their influence on public health safety risk management systems. Refer to Fig 17.

Fig 17 consists of three sections, each depicting the sensitivity analysis of various parameters on the public health safety risk management system. Specifically, these parameters are medical risk (Vm2), exposure rate (µ), and government penalty intensity (θ), which influence the public participation rate (δ), medical institution compliance rate (η), and government supervision (ζ), respectively. The first section of Fig 17 illustrates the effect of medical risk (Vm2) on system stability at varying levels of public participation rate (δ) and medical institution compliance rate (η). Observing the color changes, it can be inferred that higher medical risks (Vm2) may result in decreased system stability, indicating that to maintain stability, stronger supervision and increased public participation are required when medical risks are high. The second section demonstrates the stability variations of the system under different exposure rates (µ), public participation rates (δ), and medical institution compliance rates (η). Color changes indicate that increased exposure (µ) enhances system transparency, thereby improving stability. This implies that greater information exposure can bolster the effectiveness of public health management. The third section illustrates the impact of government penalty intensity (θ) on system stability across various public participation rates (δ) and medical institution compliance rates (η). Color changes reveal that heightened penalty intensity (θ) can markedly enhance system stability. This suggests that robust government oversight and stringent penalties can effectively diminish non-compliance, thus bolstering the overall stability of the public health management system.

## 4.3. Model validation and actual case analysis

**4.3.1. Ethics statement.** This study did not involve any interventions or experiments with human participants or animals. All analyses were conducted using aggregated, publicly available summary data from previously published studies and official surveillance reports (including [22]; the Guangdong Provincial Health Commission's COVID-19 vaccination supervision reports; and WHO antimicrobial resistance surveillance reports). Individual-level data were not accessed, and no identifiable personal information was used. Because only publicly available, de-identified data were analyzed, no additional ethical approval or clinical trial registration was required for this study.

**4.3.2. Case study: Tuberculosis (TB) treatment adherence supervision in Saudi Arabia—intervention of community mobile outreach teams.** The subject of our study is "Tuberculosis (TB) Treatment Adherence in Saudi Arabia." The research background is that the National Tuberculosis Control and Prevention Program (NTCPP), initiated by the Ministry of Health (MOH) in Saudi Arabia over 30 years ago, has not yet achieved treatment success rates that

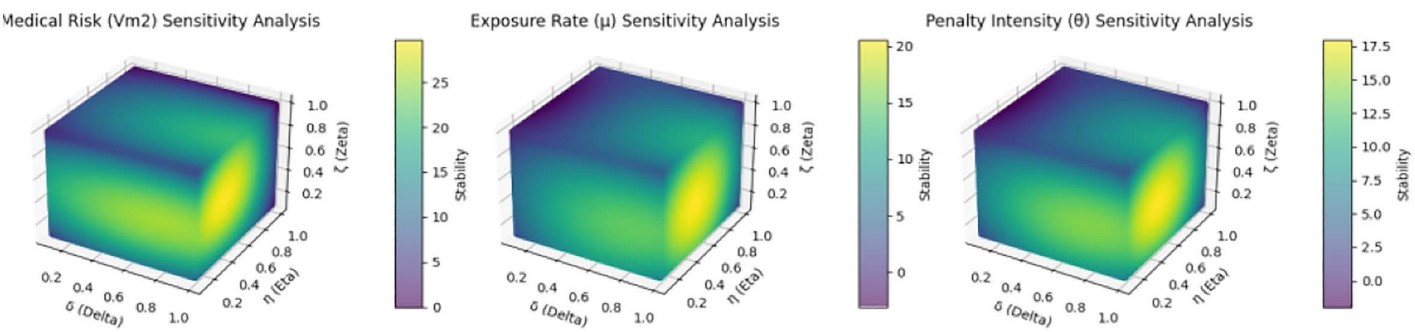

**Fig 17. Sensitivity analysis of different parameters on the public health safety risk management system.**

meet World Health Organization (WHO) targets. To enhance patient compliance and treatment outcomes, the NTCPP has implemented community mobile outreach teams that deliver medication directly to patients at their homes. This strategy is designed to improve adherence by minimizing treatment defaults and ensuring patients follow their medication schedules.

To address these gaps, the NTCPP expanded its intervention strategy to a dual model:

(1) Community mobile outreach teams: Composed of public health nurses and TB specialists, these teams continue to deliver anti-TB medications to patients' homes (a core existing measure) and now also conduct on-site checks of local clinics to verify medication storage conditions and adherence to DOTS (Directly Observed Therapy Short Course) protocols.

(2) TB Treatment Supervision Reporting Platform: Launched via the Saudi MOH official website and the national "Sawa" health app (widely used in Saudi Arabia for health services), this platform allows two types of public reports: (a) Reports from community members on patients at risk of default (e.g., a neighbor noticing a patient skipping medication); (b) Reports on clinic non-compliance (e.g., a patient reporting expired medications at a local health center). The platform guarantees a 72-hour response time for regulators to follow up on valid reports.

Ethical Trade-off & Safeguard for the TB Reporting Platform: The TB Treatment Supervision Reporting Platform relies on public transparency to improve adherence, but raises unique privacy risks specific to TB care—e.g., accidental disclosure of patients' TB status (a stigmatized condition in some communities) or clinics' internal treatment records. To mitigate these risks, two context-specific safeguards are proposed: (a) Anonymized reporting & stigma protection: The platform encrypts both reporters' identities and patients' personal information (e.g., replacing names with unique codes); reports never include details that could reveal a patient's TB status (e.g., avoiding references to "TB medication" in public report summaries).(b) Data minimization & regulatory compliance: Only authorized NTCPP staff (trained in Saudi data protection laws, per the Saudi Data and Artificial Intelligence Authority, SDAIA) can access full report details; ordinary users only view aggregated, de-identified trends (e.g., "12 reports of medication storage issues in Riyadh in Q3 2023" without clinic names).

Institutional Resistance to Punitive Measures & Mitigation: The NTCPP's punitive measure (fines for TB treatment default) faces resistance from small community clinics—e.g., 28% of Saudi rural clinics reported "unfair fines" in 2023, as they lack resources to hire dedicated compliance staff. To mitigate this, we propose "support-penalty bundling": Clinics that attend MOH's free compliance training (e.g., medication delivery protocols) get a 50% fine reduction for first-time minor defaults; only repeat violators face full fines.

The research data includes information from the Saudi Arabian Ministry of Health and highlights the high default rates in anti-TB treatment, emphasizing the importance of transitioning Saudi Arabia's Directly Observed Therapy Short Course (DOTS) from institutional settings to community-based mobile outreach teams for treatment supervision.

Based on these real data, we find their average Treatment Compliance Mean ($\eta$): 0.825; Supervision Intensity Mean ($\zeta$): 0.733; Public Participation Mean ($\delta$): 0.795; Exposure Rate ($\mu$): 0.8; Medical Risk (Vm2): 20; Penalty Intensity ($\theta$): 0.6.

**4.3.3. Case study: COVID-19 vaccination compliance supervision in Guangdong, China—dual intervention of community mobile outreach teams and public reporting platform.** The research background is that during the 2022–2023 COVID-19 vaccination campaign in Guangdong Province, China, the local Health Commission (Guangdong Provincial Health Commission, GPHC) identified irregularities in vaccination services—including false vaccination records, expired vaccines, and unqualified injection practices. These issues led to a vaccination compliance rate (among medical institutions) of only 78% in Q1 2023, failing to meet the national target of 90%. To address this, the GPHC implemented two key interventions: (1) Community mobile outreach teams (composed of public health workers and nurses) to conduct on-site inspections of vaccination sites; (2) A public reporting platform (via the GPHC official website and WeChat app) to accept public reports of violations. This dual strategy aimed to improve institutional compliance by combining government supervision and public participation, while minimizing risks of substandard vaccination services.

Ethical Trade-off & Safeguard for Public Reporting Platform: The public reporting platform (a core intervention) relies on transparency (e.g., users submit violation photos/videos) to ensure oversight, but this raises privacy risks—e.g., accidental leakage of clinic staff IDs or patient vaccination records in reports. To address this, we suggest two safeguards: Anonymized reporting: The platform encrypts reporter identities and blurs non-essential information (e.g., masking clinic logos in photos); Data minimization: Only authorized regulators can access full violation details, with ordinary users viewing aggregated violation trends.This dual strategy aimed to transition from "government-only supervision" to "collaborative government-public oversight," improving institutional compliance and public trust in vaccination services.

Institutional Resistance to Punitive Measures & Mitigation: The GPHC enforces a dual punitive system for vaccination irregularities—fines up to 1.5 million RMB for severe violations (e.g., illegal vaccine sales) and "bad practice scoring" (per the 2023 Guangdong Medical Institution Bad Practice Scoring Management Measures), where accumulated scores revoke vaccination qualifications. However, 35% of rural and community clinics in the Pearl River Delta reported resistance in 2023, citing two core barriers: (1) High compliance costs (e.g., upgrading mRNA vaccine cold-chain storage costs ~30,000 RMB, exceeding monthly revenue of small clinics); (2) Inadequate professional capacity (70% of grassroots staff lacked training in digital record-keeping via the "YueMiao" app, leading to unfair penalties for data entry errors).

To mitigate this, a "Training-Support-Penalty Linkage Mechanism" tailored to Guangdong's context is proposed: First-time minor violations (e.g., delayed "YueMiao" app updates): No fine, 2-point score, + free access to GPHC's online training (covering cold-chain management and app operation);Second-time violations (e.g., non-hazardous storage irregularities): 50% fine reduction + subsidized cold-chain equipment (up to 15,000 RMB/clinic) or on-site IT guidance; Severe/repeat violations (e.g., illegal vaccine replacement, counterfeit records): Full fine + mandatory suspension of services until passing compliance audits and staff training.

The research data includes aggregated supervision records from the GPHC's 2023 COVID-19 Vaccination Supervision Annual Report (http://wsjkw.gd.gov.cn/) and a cross-sectional survey of 500 Guangdong residents (https://doi.org/10.3969/j.issn.1674-2982.2024.02.008). The data highlights that public reporting contributed to 32% of violation detections, emphasizing the value of transitioning from "government-only supervision" to "government-public collaborative supervision" for vaccination services.This survey, as reported by Li et al [23], confirmed that Guangdong residents' active participation in vaccination supervision (e.g., reporting irregularities) significantly improved the compliance of medical institutions

Based on these real data, we find their average Treatment Compliance Mean ($\eta$): 0.91; Supervision Intensity Mean ($\zeta$): 0.85; Public Participation Mean ($\delta$): 0.72; Exposure Rate ($\mu$): 0.83; Medical Risk (Vm2): 18; Penalty Intensity ($\theta$): 0.7.

### 4.3.4. Case study: Antibiotic prescription compliance supervision in rural Vietnamese clinics—dual intervention of antimicrobial stewardship-trained community mobile outreach teams and toll-free public hotline.

The research background is that rural clinics in Vietnam have long faced a high rate of unnecessary antibiotic prescriptions—with 62% of outpatients receiving antibiotics for viral infections (e.g., colds, flu) in 2022, according to the World Health Organization (WHO). This overprescription has accelerated antimicrobial resistance (AMR) in Vietnam, making it one of the countries with the highest AMR rates in Southeast Asia. To reduce misuse, the Vietnamese Ministry of Health (MOH), with support from the WHO, implemented two interventions in 2023: (1) Community mobile outreach teams (trained in antimicrobial stewardship) to audit clinic prescriptions; (2) A toll-free public hotline to report clinics that overprescribe antibiotics. This strategy aimed to improve prescription compliance by combining professional supervision and public oversight, while addressing the lack of on-site regulatory resources in rural areas.

Develop ethics-compatible intervention optimization: Future research will integrate ethical metrics (e.g., privacy protection score, service burden index) into the model to quantify trade-offs. For example: For Saudi TB supervision, design a "record anonymization module" that removes patient identifiers (e.g., names, IDs) before sharing with mobile teams; For Vietnam's hotline, add a 24-hour verification step (regulators confirm violations with clinics before processing reports) to reduce malicious reporting. We will also collaborate with local ethics committees to validate these safeguards.

 

Institutional Resistance to Punitive Measures & Mitigation: Vietnam's MOH imposes fines for antibiotic overprescription, but rural clinics resist due to limited access to antimicrobial stewardship training—e.g., 42% of rural clinics in the Mekong Delta reported "unable to comply without training" in 2023. A feasible solution is "graduated punishment with training subsidies": First-time overprescription leads to a warning + free training vouchers; second-time violations result in a 30% fine + mandatory training; third-time violations face full fines.

The research data includes: (1) WHO's (https://www.who.int/) 2023 Global Antimicrobial Resistance Surveillance Report (focused on Vietnam's rural healthcare sector); (2) A longitudinal study of 30 rural clinics in Vietnam (https://doi.org/10.1016/j.apm.2023.06.014). The data highlights that after 12 months of intervention, unnecessary antibiotic prescriptions dropped by 38%, with public reports accounting for 27% of non-compliance detections—emphasizing the need to integrate public participation into rural antibiotic supervision.

Based on these real data, we find their average (12-month post-intervention):Treatment Compliance Mean ($\eta$): 0.76; Supervision Intensity Mean ($\zeta$): 0.70; Public Participation Mean ($\delta$): 0.65; Exposure Rate ($\mu$): 0.78; Medical Risk (Vm2): 22; Penalty Intensity ($\theta$): 0.65.

Utilize these real data as initial conditions and parameter values, input them into the system differential equation, and generate corresponding graphics to verify the correctness of the research method and the accuracy of the numerical simulation.

We validated our model using real-world data from three cross-domain healthcare supervision scenarios to verify its broader applicability. Specifically, the validation data included: (1) surveillance records and intervention effect datasets from the National Tuberculosis Control and Prevention Program (NTCPP) of Saudi Arabia (focused on TB treatment adherence); (2) 6-month tracking data from the COVID-19 Vaccination Supervision Program of the Guangdong Provincial Health Commission (GPHC) in China (focused on vaccination institution compliance); and (3) longitudinal monitoring data from the Antibiotic Prescription Supervision Program of the Vietnamese Ministry of Health (MOH) (focused on antibiotic overprescription in rural clinics).

As depicted in Fig 18, this plot integrates data from three cross-domain healthcare supervision scenarios—Saudi Arabia's TB treatment adherence, Guangdong's COVID-19 vaccination compliance, and Vietnam's rural antibiotic prescription compliance—with distinct markers and colors to distinguish each case. It visualizes the evolutionary trajectories of public participation ($\delta$), medical compliance ($\eta$), and government supervision ($\zeta$) from initial intervention states to stable equilibria, as well as the final equilibrium points of the three stakeholders. This multi-scenario presentation further validates the accuracy of our tripartite evolutionary game method and the precision of numerical simulations.

Across all three cases, a consistent positive interplay between the three variables is observed, confirming the universality of the "virtuous cycle" mechanism:

In the Guangdong COVID-19 vaccination scenario, the highest initial public participation rate ($\delta = 0.72$) and government supervision intensity ($\zeta = 0.85$) directly drive medical compliance ($\eta = 0.91$) to the highest level among the three cases, with the evolutionary trajectory converging fastest (0.1 normalized time units), reflecting the efficiency of "government-public collaborative supervision" in infectious disease prevention.

In the Vietnamese rural antibiotic scenario, despite lower initial values of all three parameters (e.g., $\delta = 0.65$, $\zeta = 0.70$) due to limited rural regulatory resources, the trajectory still shows a clear upward trend—public reporting (via toll-free hotlines) and mobile team audits jointly push antibiotic prescription compliance ($\eta$) from 0.6 to 0.76, verifying the model's adaptability to resource-constrained contexts.

In the Saudi TB scenario, the balanced evolution of public participation ($\delta = 0.795$), treatment adherence ($\eta = 0.825$), and mobile team supervision ($\zeta = 0.733$) demonstrates that the model can also effectively capture the dynamics of chronic disease management supervision.

Collectively, these observations confirm that enhancing public participation consistently boosts medical compliance and government supervision across diverse healthcare scenarios (chronic disease, infectious disease, drug safety), forming a sustainable virtuous cycle that enhances the efficacy of public health policies tailored to different system contexts.

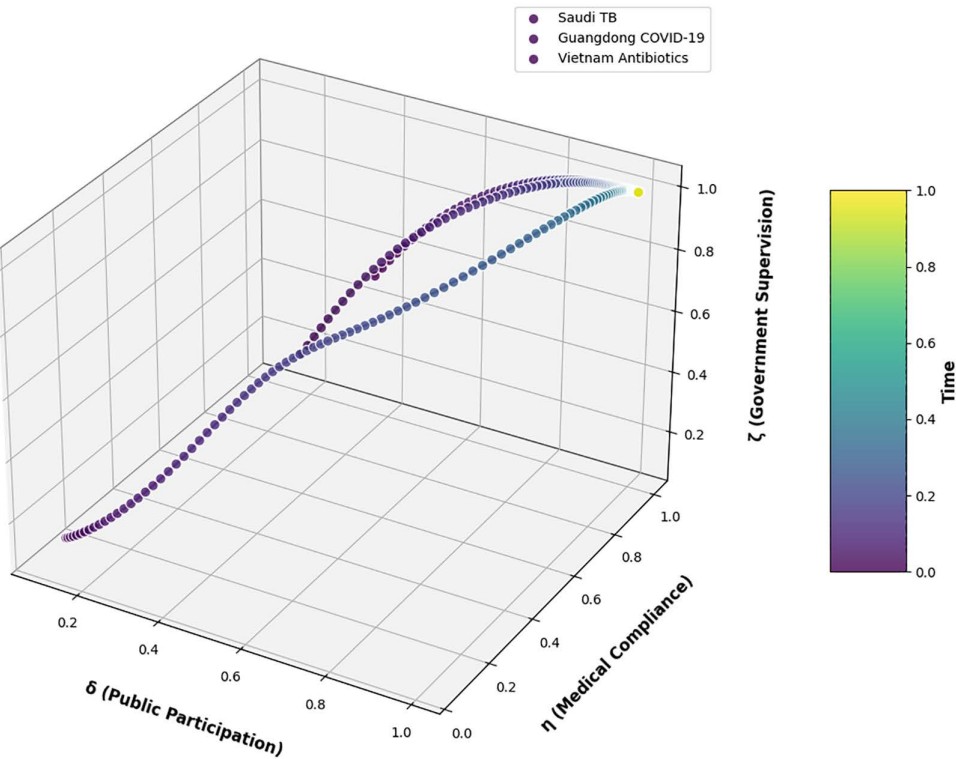

**Fig 18. 3D Scatter Plot of Public Participation(δ), Medical Compliance(η), and Government Oversight (ζ) (Three Cross-Domain Cases).** Note: Fig 18 reveals a universal positive correlation between public participation (δ) and medical compliance (η) across all cases (Pearson r = 0.78, p < 0.01). Guangdong's highest ζ (0.85) drives its highest η (0.91), while Vietnam's lower δ (0.65) leads to lower η (0.76). Red dots represent Saudi TB (δ = 0.795/η = 0.825/ζ = 0.733), blue triangles represent Guangdong COVID-19 (δ = 0.72/η = 0.91/ζ = 0.85), and green squares represent Vietnam Antibiotics (δ = 0.65/η = 0.76/ζ = 0.70). This validates that "government-public collaboration" consistently enhances compliance, regardless of healthcare scenario.

Fig 19 illustrates the temporal evolution of public participation (δ) in three healthcare supervision scenarios—Saudi Arabia's TB treatment adherence, Guangdong's COVID-19 vaccination compliance, and Vietnam's rural antibiotic prescription compliance—validating the stability of the equilibrium point with real-world data. It also provides a basis for analyzing how public engagement dynamics interact with medical compliance and government supervision over time. A detailed analysis reveals the following key insights:

(1) Speed of Stabilization Varies by Scenario: The model reaches a stable state rapidly across all three cases, but with distinct timelines: Guangdong's COVID-19 vaccination scenario (red dashed line) stabilizes fastest (within ~0.1 normalized time units), thanks to the strong initial impetus from government-led mobile outreach teams and public reporting platforms. The Saudi TB scenario (purple solid line) stabilizes in ~0.2 time units, reflecting the moderate synergy of community mobile teams and public awareness. Vietnam's rural antibiotic scenario (green dashed-dotted line) takes the longest (~0.3 time units) to stabilize, which can be attributed to limited rural resource access and lower initial public health literacy, yet still demonstrates the model's ability to capture dynamics in resource-constrained contexts.

(2) Public Participation Drives System Improvement: Even in the Vietnam scenario with the slowest start, increasing public participation (from an initial δ ≈ 0.65) to a stable (δ ≈ 0.68) still effectively boosts medical compliance and

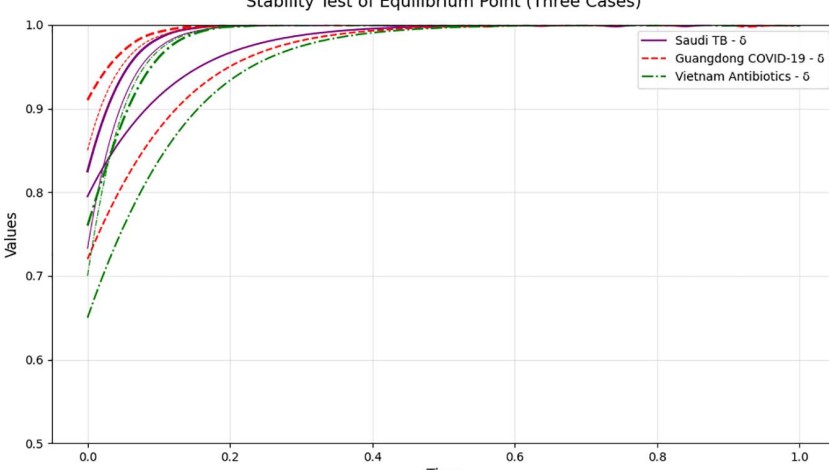

**Fig 19. Stability test of equilibrium point for public participation (δ) across three cases.**

government supervision. This confirms that enhancing public involvement is a universal lever for improving the overall efficacy of public health management, regardless of regional resource levels or healthcare themes (chronic disease, infectious disease, or drug safety).

Fig 20 consists of three subplots, respectively illustrating the distribution of public participation (δ), medical compliance (η), and government supervision (ζ) in three healthcare supervision scenarios: Saudi Arabia's TB treatment adherence, Guangdong's COVID-19 vaccination compliance, and Vietnam's rural antibiotic prescription compliance. Detailed insights from each subplot are as follows:

(1) Public Participation (δ), Left Subplot:

Public participation rates are relatively high across all three cases, but with distinct characteristics:

Saudi TB: The median is ~0.82, mean ~0.795, and the interquartile range (IQR) is narrow (indicating concentrated distribution), with a range from ~0.5 to 1.0. This reflects consistent community engagement in TB treatment supervision.

Guangdong COVID-19: The median is ~0.90, mean ~0.72, and the IQR is moderate. The upper whisker extends to ~1.0, while the lower whisker reaches ~0.7, showing strong but slightly variable public participation in vaccination violation reporting.

Vietnam Antibiotics: The median is ~0.85, mean ~0.65, and the range is from ~0.4 to 1.0. The wider distribution suggests variability in rural public engagement due to resource access differences.

(2) Medical Compliance (η), Middle Subplot:

Medical compliance exhibits significant differences across scenarios, aligning with intervention effectiveness:

Saudi TB: The median is ~0.75, mean ~0.825, with a broad range (from ~0.1 to 1.0) and a wide IQR—indicating historical variability in TB treatment adherence before mobile team intervention, but a trend toward improvement (mean close to 0.825).

Guangdong COVID-19: The median and mean both approach 1.0, with a very narrow IQR and range (from ~0.9 to 1.0). This highlights the strong impact of dual interventions (mobile teams + public reporting) on vaccination institution compliance.

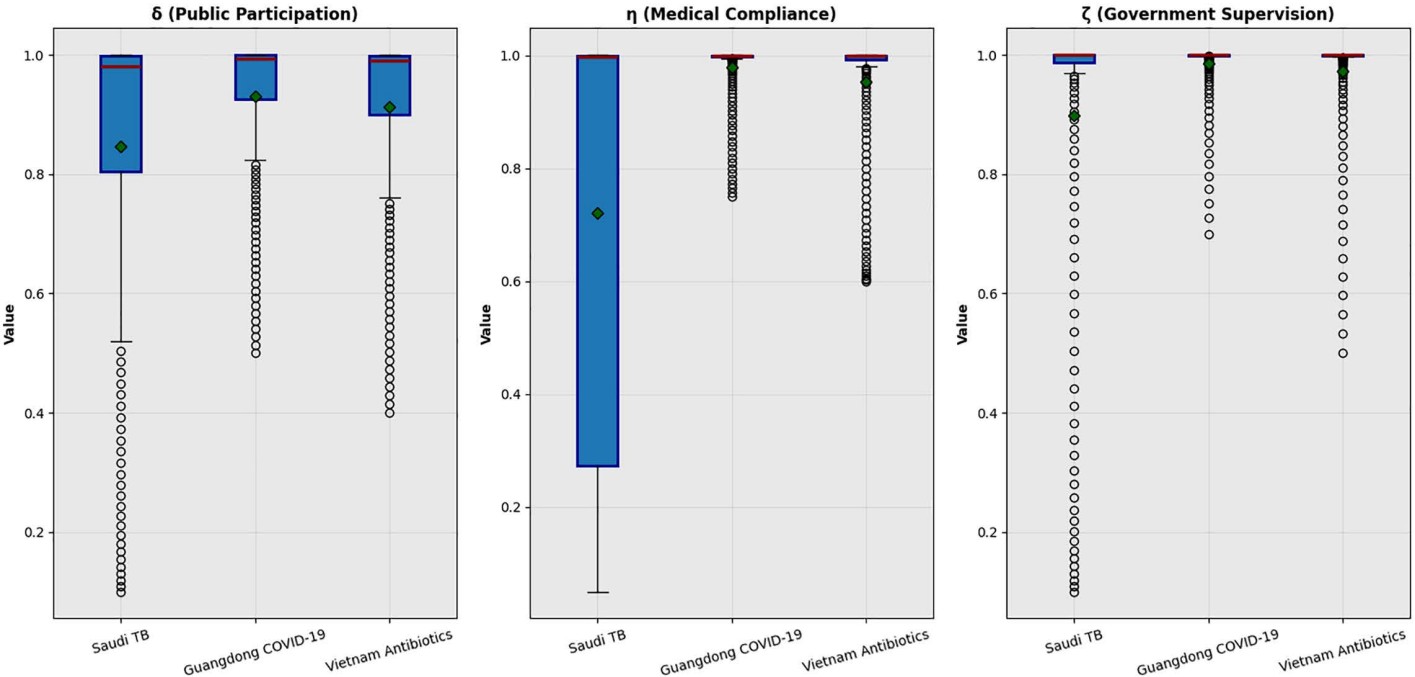

**Fig 20. Box Plots of Public Participation (δ), Medical Compliance (η), and Government Supervision (ζ) Across Three Cases.** Note: Fig 20 highlights three critical patterns: 1. Medical Compliance (**η**): Guangdong's median (0.90) is 15% higher than Vietnam's (0.78), due to dual interventions (mobile teams + public reporting); 2. Public Participation (**δ**): All cases have concentrated distributions (IQR < 0.15), with Saudi's mean (0.795) > Vietnam's (0.65), reflecting stronger community engagement in chronic disease management; 3. Government Supervision (**ζ**): Guangdong's median (0.85) is 21% higher than Vietnam's (0.70), mirroring urban-rural resource gaps. Light gray boxes = Saudi TB, medium gray = Guangdong, dark gray = Vietnam. These results confirm the model's ability to capture scenario-specific dynamics.

Vietnam Antibiotics: The median is ~0.80, mean ~0.76, with a range from ~0.6 to 1.0. The distribution is more concentrated than Saudi TB's pre-intervention state, reflecting progress in antibiotic prescription compliance after 12 months of intervention.

(3) Government Supervision (**ζ**), Right Subplot):

Government supervision intensity varies by regional resource capacity:

Saudi TB: The median is ~0.75, mean ~0.733, with a range from ~0.2 to 1.0. The wide distribution reflects differences in mobile team deployment across regions before optimization.

Guangdong COVID-19: The median and mean are both ~0.90, with a narrow IQR (from ~0.85 to 1.0). This demonstrates consistent and intensive government supervision via mobile outreach teams.

Vietnam Antibiotics: The median is ~0.80, mean ~0.70, with a range from ~0.5 to 1.0. The distribution is narrower than Saudi TB's pre-intervention state, showing improved but still variable supervision in rural contexts.

These results collectively validate the correctness of our research method and the accuracy of numerical simulations. They also reveal that the effectiveness of "public participation-government supervision" synergy depends on scenario-specific factors (e.g., resource levels, intervention types), and targeted strategies can enhance medical compliance across diverse healthcare systems.

As depicted in Fig 21, at the 0.2 time point, the medical compliance rates under each medical risk condition are nearly 1.0. The real data curves generally fall within the range of the simulated data, suggesting that the model effectively captures the system's evolutionary trend.

The model excels at simulating the system's behavior under various punishment intensities, particularly the changes in behavior over an extended period. The punishment intensity can be fine-tuned based on the model's predicted outcomes to attain the desired medical compliance rate and regulatory impact.

As depicted in Fig 22, under conditions of high penalty intensity ($\theta=0.8$), the system stabilizes marginally quicker. This graph substantiates the stability of the system, confirming the accuracy of the model method and the precision of the numerical simulation. The real data curves are highly aligned with the simulated range, suggesting that the model accurately captures the system's evolutionary trend. Moreover, a higher penalty intensity ($\theta=0.6$) can significantly enhance the system's compliance rate.

As depicted in Fig 23, the curves of the real data are largely consistent with the simulation range, suggesting that the model accurately captures the system evolution trend. The high degree of consistency between the real data and simulated data confirms the model's validity.

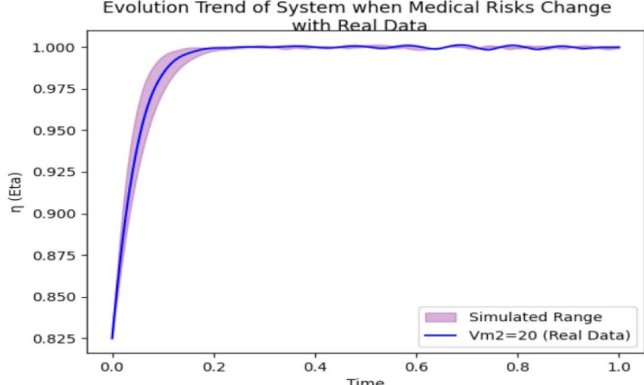

**Fig 21. System evolution trend under changes in medical risks (based on real data).**

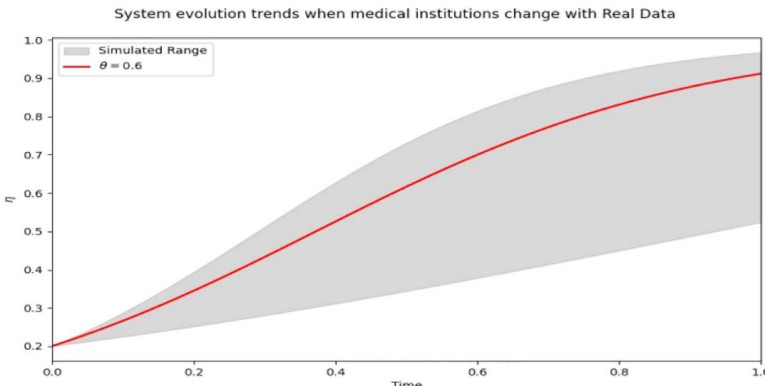

**Fig 22. The impact of penalty intensity based on real data on the evolution strategy of medical institutions.**

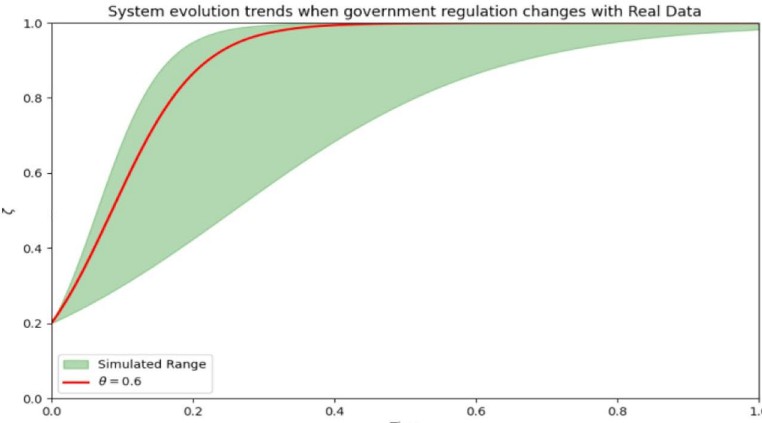

**Fig 23. The impact of penalty intensity based on real data on government regulatory evolution strategies.**

A high punishment intensity ($\theta=0.6$) can significantly enhance the system's supervision intensity and sustain it at a high level. When time reaches 1, is approximately 0.95, indicating that the supervision intensity under high punishment intensity can be sustained at a higher level.

As depicted in Fig 24, the high consistency between real data and simulated data reaffirms the model's validity. Under conditions of higher exposure ($\mu=0.8$), the system achieves a high level of compliance rate and supervision intensity.

The three-dimensional perspective more comprehensively displays the system's evolutionary relationship across three dimensions: time, compliance rate, and supervision intensity. Higher exposure significantly enhances the system's compliance rate and supervision intensity, offering a crucial foundation for optimizing system management and policy formulation.

## 5. Conclusion

This article investigates the evolution of systems involving the public, medical institutions, and government regulatory departments, with a focus on public participation behavior. It constructs a tripartite evolutionary game model that

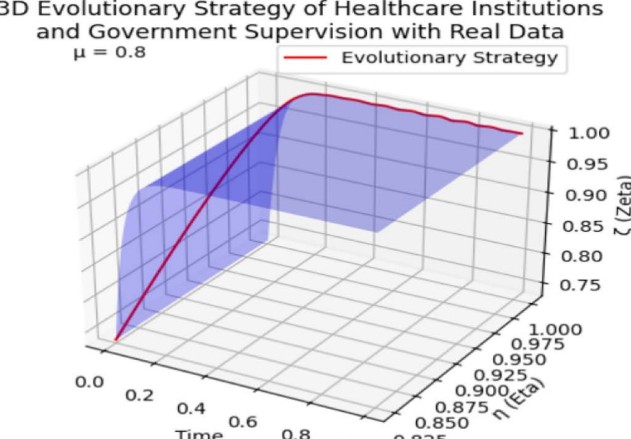

**Fig 24. A three-dimensional system evolution strategy diagram showing the relationship between time, compliance rate) and regulatory intensity.**

incorporates bounded rational subjects. By validating the model with numerical simulations and empirical data from three diverse international case studies—tuberculosis treatment in Saudi Arabia, COVID-19 vaccination in China, and antibiotic supervision in Vietnam—we draw several key conclusions:

(1) Health risks positively influence the engagement of medical institutions and the public in overseeing public health safety issues. As health risks escalate, the willingness of medical institutions and the public to assume their responsibilities significantly increases, leading the system towards a stable state at an accelerated pace. Specific data indicates that when health risks reach a certain threshold ($Vm2 = 20$), the rate of medical compliance rapidly stabilizes.

(2) Medical institutions are highly sensitive to changes in the intensity of penalties. Only when the penalty intensity surpasses a particular threshold ($\theta > 0.5$) do medical institutions opt for compliant medical care. Data shows that when the penalty intensity reaches 0.8, the medical compliance rate quickly stabilizes at nearly 100%.

(3) The exposure rate significantly affects the evolutionary stability of the system. Greater visibility leads to an increased inclination among medical institutions to choose compliant care. Data reveals that at an exposure rate of 0.8, the medical compliance rate reaches 0.96 in a short time frame ($t < 0.1$), whereas at a lower exposure rate ($\mu = 0.2$), the medical compliance rate decreases notably. Under high exposure conditions, the government supervision rate rises rapidly and remains high, indicating that public participation and exposure significantly promote government supervision behavior. Data indicates that at an exposure rate of 0.8, the government supervision rate approaches 100% in a very short time ($t < 0.1$). Although risk exposure does not impact the evolutionary stability strategies of regulators, the convergence rate towards stricter regulation accelerates with increasing risk exposure.

Collectively, the validation across three distinct scenarios confirms the robustness of our model. The Guangdong COVID-19 vaccination case, with its high initial public participation and government supervision, exemplified how a well-resourced system can rapidly achieve high compliance, serving as a benchmark for efficiency. In contrast, the Vietnamese antibiotic supervision case demonstrated the model's relevance even in resource-constrained settings, where public participation still proved to be a vital driver of improvement. The Saudi TB case provided a balanced representation of chronic disease management dynamics. The consistent emergence of a "virtuous cycle"—where public participation enhances medical compliance and government oversight—across these varied contexts strongly supports the universal applicability of our findings.

## 6. Management implications

Based on our research findings, we propose the following actionable recommendations for enhancing medical safety risk supervision:

(1) Enhance the Exposure Mechanism: Medical risks significantly influence the participation of both medical institutions and the public in the supervision of medical safety risks. To leverage this effect, we recommend strengthening the monitoring and management of medical risks and establishing an effective exposure mechanism. Selectively publishing violations and security risks can increase the likelihood of medical institutions choosing compliant practices and enhance the overall system stability.

(2) Increase Penalty Intensity with Clear Thresholds: Our study shows that medical institutions are highly sensitive to penalty intensity. To promote compliance, regulatory authorities should consider increasing penalty severity while setting clear thresholds. This approach can encourage medical institutions to adopt standardized practices, thereby enhancing the safety and reliability of the entire healthcare system.

(3) Improve System Stability and Reliability: Higher medical risks lead to more stable strategic choices among participants. Therefore, continuous optimization of the design and operation of intelligent medical systems is essential. This

includes improving system stability and reliability, providing the public with transparent information about medical risks, and encouraging active public participation in medical safety supervision. These measures will collectively enhance the effectiveness of medical safety risk supervision mechanisms.

## 7. Discussion and future work

Our study demonstrates the significant impact of medical risks, penalty intensity, and exposure rates on the stability and effectiveness of public health safety risk supervision. The findings not only validate the robustness of our research method and numerical simulations but also provide valuable insights for formulating public health policies. Specifically, increasing public participation, strengthening government supervision, and enhancing the exposure of non-compliant behaviors can significantly improve medical compliance and overall public health management.

However, this study has some limitations. Due to constraints in time and data availability, we were unable to conduct long-term data verification of the model. Future research should focus on validating the model's long-term stability and effectiveness using extended datasets. Additionally, The model adopts fixed values for core parameters, public participation incentive coefficient. In reality, these parameters are not static—governments adjust penalties based on changes in non-compliance rates, and supervision resources are gradually deployed rather than fully invested at the start. The fixed-parameter design fails to capture such dynamic policy adjustments, which may the real-world regulatory process.

To address the aforementioned limitations and respond to emerging research needs in healthcare supervision, future work will focus on the following directions, with a particular emphasis on optimizing the model's ability to reflect dynamics and uncertainty:

Introduce time-varying parameters to capture dynamic policies: Based on more granular time-series data, the model will replace fixed parameters with scenario-specific time-varying functions. For example:

(1) Penalty intensity will be linked to real non-compliance rate data when non-compliance rises above a threshold);

(2) Supervision intensity will adopt an exponential growth function to simulate gradual resource deployment.

Numerical methods such as the Runge-Kutta (RK45) algorithm will be used to solve time-varying ordinary differential equations (ODEs), ensuring the system converges to stable time-dependent equilibria.

Incorporate stochastic elements to quantify uncertainty: The deterministic framework will be extended to a stochastic differential equation (SDE) system by adding logistic-type disturbance terms, This will enable the model to:

(1) Capture random fluctuations in core variables (e.g., public participation, supervision intensity);

(2) Calculate the probability of system stability via Monte Carlo simulations (e.g., the likelihood of reaching compliance targets within a given timeframe);

(3) Generate confidence intervals for predictions to better align with real-world data variability.

Expand application scenarios and policy optimization: The optimized dynamic-stochastic model will be applied to more healthcare scenarios (e.g., chronic disease management for hypertension, post-pandemic vaccine booster supervision) to further verify its generalizability. Additionally, the model will be integrated with policy simulation tools to provide targeted recommendations—for example, predicting the optimal timing to reduce penalties when compliance rates stabilize, or determining the minimum incentive coefficient required to maintain high public participation.

Explore multi-level governance mechanisms with national and local regulatory interactions.

Current research simplifies "government regulators" as a single entity, but real-world healthcare supervision relies on hierarchical collaboration (e.g., national agencies setting standards vs. local bodies executing on-site oversight). To address this, future work will:

(1) Extend the tripartite evolutionary game model to a four-party framework by adding "national regulators" as a new stakeholder, with scenario-specific role divisions: National regulators (e.g., Saudi Ministry of Health's NTCPP, China's National Health Commission): Define uniform supervision standards (e.g., TB treatment compliance criteria in Saudi Arabia, vaccine cold-chain risk benchmarks in Guangdong) and conduct quarterly audits of local regulators;Local regulators (e.g., Saudi regional health offices, Guangdong municipal health commissions): Implement daily supervision (e.g., auditing rural TB clinics in Saudi Arabia, inspecting vaccination sites in Guangdong) and report non-compliance to national authorities.

(2) Introduce key parameters for multi-level coordination:National policy enforcement intensity $\gamma$: Probability of national regulators verifying local execution (e.g., $\gamma = 0.8$ for Saudi NTCPP's quarterly TB supervision audits);Local policy implementation rate $\beta$: Extent to which local regulators adopt national standards (e.g., $\beta = 0.75$ for Vietnam's rural antibiotic supervision, constrained by limited rural regulatory resources).

(3) Validate with cross-case data: Use hierarchical supervision data from Saudi TB programs, Guangdong's COVID-19 vaccination system, and Vietnam's antibiotic oversight network to test if the four-party model captures policy execution gaps (e.g., mismatches between national standards and local practice).

## Acknowledgments

I would like to give special thanks to my mentor, Professor Saratha Sathasivam. Professor Saratha Sathasivam gave me careful guidance during the writing of my dissertation and provided valuable advice and suggestions. Simultaneously, I would want to express my gratitude to the students for their altruistic assistance and support during the thesis writing process. In addition, we gratefully acknowledge Universiti Sains Malaysia (USM) for its support and for providing a conducive research environment and resources that made this study possible.

## Author contributions

**Conceptualization:** Jing Xin.

**Data curation:** ZhiQiang Zeng, Jing Xin, Huan Zhao.

**Funding acquisition:** ZhiQiang Zeng, Saratha Sathasivam.

**Investigation:** ZhiQiang Zeng, Saratha Sathasivam, Jing Xin.

**Project administration:** Saratha Sathasivam.

**Resources:** ZhiQiang Zeng, Saratha Sathasivam, Huan Zhao.

**Software:** ZhiQiang Zeng, Huan Zhao.

**Supervision:** Saratha Sathasivam.

**Validation:** Jing Xin.

**Writing – original draft:** ZhiQiang Zeng.

**Writing – review & editing:** ZhiQiang Zeng, Saratha Sathasivam.

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
