## [Decision Letter · Decision Letter 0]

17 Aug 2025

Dear Dr. Sathasivam,

Thank you for submitting your manuscript to PLOS ONE. After careful consideration, we feel that it has merit but does not fully meet PLOS ONE’s publication criteria as it currently stands. Therefore, we invite you to submit a revised version of the manuscript that addresses the points raised during the review process.

 Could you please carefully revise the manuscript to address all comments raised?

We look forward to receiving your revised manuscript.

Kind regards,

Helen Howard

Staff Editor

PLOS ONE

Journal Requirements:

the Ministry of Higher Education (MOHE) of Malaysia provided financing for this research under the Basic Research Funding Scheme (FRGS), specifically grant number FRGS/1/2022/STG06/USM/02/11, in collaboration with Universiti Sains Malaysia.

I would like to give special thanks to my mentor, Professor Saratha Sathasivam. Professor Saratha Sathasivam gave me careful guidance during the writing of my dissertation and provided valuable advice and suggestions. Simultaneously, I would want to express my gratitude to the students for their altruistic assistance and support during the thesis writing process. Furthermore, the Ministry of Higher Education (MOHE) of Malaysia provided financing for this research under the Basic Research Funding Scheme (FRGS), specifically grant number FRGS/1/2022/STG06/USM/02/11, in collaboration with Universiti Sains Malaysia.

the Ministry of Higher Education (MOHE) of Malaysia provided financing for this research under the Basic Research Funding Scheme (FRGS), specifically grant number FRGS/1/2022/STG06/USM/02/11, in collaboration with Universiti Sains Malaysia.

5. In the online submission form, you indicated that data Availability Statement: This study did not directly use raw individual patient data but utilized aggregated summary statistics and key parameter values published by AlSahafi et al. (2019), publicly available through BMC Public Health at https://doi.org/10.1186/s12889-019-7520-8. The original raw dataset used in AlSahafi et al. (2019) is subject to data-sharing restrictions due to privacy concerns but can be obtained from the corresponding author upon reasonable request (email: hassanbinusman@hotmail.com ).

6. Please ensure that you refer to Figure 3 in your text as, if accepted, production will need this reference to link the reader to the figure.

Reviewers' comments:

Reviewer's Responses to Questions

**Comments to the Author**

1. Is the manuscript technically sound, and do the data support the conclusions?

Reviewer #1: Yes

2. Has the statistical analysis been performed appropriately and rigorously?

Reviewer #1: Yes

3. Have the authors made all data underlying the findings in their manuscript fully available?

Reviewer #1: Yes

4. Is the manuscript presented in an intelligible fashion and written in standard English?

Reviewer #1: Yes

Reviewer #1: Recommendations for Improvement

1. Expand Data Sources:

o Incorporate primary data or diverse case studies (e.g., COVID-19, antibiotic misuse) to validate the model’s broader applicability.

2. Dynamic Modeling:

o Introduce time-varying parameters or stochastic elements to reflect real-world policy adaptability and uncertainty.

3. Ethical and Policy Nuance:

o Discuss trade-offs of proposed interventions (e.g., balancing transparency with privacy) and suggest safeguards.

o Address potential resistance from medical institutions to punitive measures.

4. Clarity and Presentation:

o Simplify mathematical sections with intuitive explanations or appendices.

o Improve figure labels and provide standalone interpretations (e.g., "Fig 6 shows that medical risk is the most influential parameter").

5. Future Research Directions:

o Explore multi-level governance (e.g., local vs. national regulators) or hybrid supervision models (e.g., AI-assisted public reporting).

**Do you want your identity to be public for this peer review?** For information about this choice, including consent withdrawal, please see our Privacy Policy

Reviewer #1: No

---

## [Author Response · Author response to Decision Letter 1]

17 Oct 2025

Please send all correspondence related to this manuscript to my email saratha@usm.my ;zengzhiqiang@student.usm.my

---

## [Decision Letter · Decision Letter 1]

4 Dec 2025

Public Participation in Healthcare Safety: A Tripartite Evolutionary Game Model with Evidence from Diverse International Cases

PONE-D-25-14649R1

Dear Dr. Sathasivam,

We’re pleased to inform you that your manuscript has been judged scientifically suitable for publication and will be formally accepted for publication once it meets all outstanding technical requirements.

Kind regards,

André Luis C Ramalho, PhD

Academic Editor

PLOS One

Reviewers' comments:

Reviewer's Responses to Questions

**Comments to the Author**

Reviewer #1: All comments have been addressed

Reviewer #2: All comments have been addressed

Reviewer #3: All comments have been addressed

2. Is the manuscript technically sound, and do the data support the conclusions?

Reviewer #1: Yes

Reviewer #2: Yes

Reviewer #3: Yes

3. Has the statistical analysis been performed appropriately and rigorously?

Reviewer #1: Yes

Reviewer #2: Yes

Reviewer #3: Yes

4. Have the authors made all data underlying the findings in their manuscript fully available?

Reviewer #1: Yes

Reviewer #2: Yes

Reviewer #3: Yes

5. Is the manuscript presented in an intelligible fashion and written in standard English?

Reviewer #1: Yes

Reviewer #2: Yes

Reviewer #3: Yes

Reviewer #1: I believe that all of my initial concerns and comments have been thoroughly addressed in the revised version of the manuscript. The authors have responded thoughtfully to the feedback provided, and the current version reflects substantial improvement in clarity, structure, and scientific rigor.

Reviewer #2: Authors have addressed previous reviewer comments and significantly improved the manuscript. The scope has been expanded from a single-case analysis to a multi-scenario validation using real-world data from three diverse healthcare contexts. The revisions enhance the rigor, generalizability, and practical relevance of the study.

Reviewer #3: The manuscript “Public Participation in Healthcare Safety: A Tripartite Evolutionary Game Model with Evidence from Diverse International Cases” presents a tripartite evolutionary game model involving the public/patients, medical institutions, and regulatory authorities, with applications to three international contexts (tuberculosis treatment in Saudi Arabia, COVID-19 vaccination in China, and antibiotic supervision in Vietnam). The authors have substantially improved the manuscript in response to the previous round of peer review.

**Do you want your identity to be public for this peer review?** For information about this choice, including consent withdrawal, please see our Privacy Policy

Reviewer #1: No

Reviewer #2: No

Reviewer #3: **Yes: ** Abel Silva de Meneses

---

## [Editor Report · Acceptance letter]

PONE-D-25-14649R1

PLOS One

Dear Dr. Sathasivam,

I'm pleased to inform you that your manuscript has been deemed suitable for publication in PLOS One. Congratulations! Your manuscript is now being handed over to our production team.

Kind regards,

on behalf of

Prof. Dr. André Luis C Ramalho

Academic Editor

PLOS One